# FACT: Fine-grained Across-variable Convolution for multivariate Time-series forecasting

**Huiqiang Wang, Jieming Shi**\*, **Qing Li**
The Hong Kong Polytechnic University
`huiqiang.wang@connect.polyu.hk`, `jieming.shi@polyu.edu.hk`
`csqli@comp.polyu.edu.hk`

## ABSTRACT

Modeling the relationships among variables has become increasingly important, particularly in high-dimensional multivariate time series forecasting tasks. However, most existing methods primarily focus on capturing coarse-grained correlations between variables, overlooking a finer and more dynamic aspect: the variable interactions often manifest differently as time progresses. To address this limitation, we propose **FACT**, an **F**ine-grained **A**cross-variable **C**onvolution architecture for multivariate **T**ime series forecasting that explicitly models fine-grained variable interactions from both the time and frequency domains. Technically, we introduce a depth-wise convolution block DConvBlock, which leverages a depth-wise convolution architecture with channel-specific kernels to model dynamic variable interactions at each granularity. To further enhance efficiency, we reconfigure the original one-dimensional variables into a two-dimensional space, reducing the variable distance and the required model layers. Then DConvBlock incorporates multi-dilated 2D convolutions with progressively increasing dilation rates, enabling the model to capture fine-grained and dynamic variable interactions while efficiently attaining a global reception field. Extensive experiments on twelve benchmark datasets demonstrate that FACT not only achieves state-of-the-art forecasting accuracy but also delivers substantial efficiency gains, significantly reducing both training time and memory consumption compared to attention mechanism. The code is available at `https://github.com/wanghq21/FACT`.

## 1 INTRODUCTION

Multivariate time series forecasting plays a pivotal role in a wide range of real-world applications, such as weather prediction (Wu et al., 2023b; Bi et al., 2023), traffic management (Yin et al., 2022), financial market analysis (Liu et al., 2024), disease surveillance (Matsubara et al., 2014), and energy system optimization (Qian et al., 2019). In such scenarios, multiple variables interact in intricate and evolving ways, exhibiting correlations, causal relations, covariances, and temporal lags in the time domain, as well as resonance, phase synchronization, and frequency-dependent coupling in the frequency domain. Accurately modeling these across-variable dependencies is therefore critical to improving forecasting performance.

A wide range of approaches have been developed to capture variable dependencies, including attention-based architectures (e.g., iTransformer (Liu et al., 2024), TimeXer (Yu et al., 2024), Crossformer (Zhang & Yan, 2023)), clustering-based approaches (e.g., DUET (Qiu et al., 2025), CCM (Chen et al., 2024a)), and CNN-based models (e.g., ModernTCN (Luo & Wang, 2024)). While effective, most of these models treat multivariate representations as indivisible wholes, thereby overlooking fine-grained and temporally evolving interactions among individual variables.

In practice, such interactions are highly dynamic. For example, the correlation between traffic flows on two adjacent streets may be strong during weekday commuting hours but weak on weekends, when travel patterns become more random. These dynamics often exhibit locality and periodicity,

---

\*Corresponding Author.

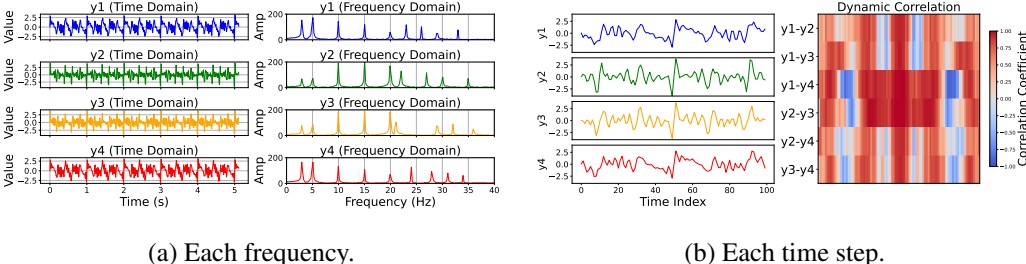

(a) Each frequency.          (b) Each time step.

Figure 1: The dynamic interaction at each granularity.

underscoring the importance of modeling variable dependencies at finer granularity and with temporal sensitivity. Furthermore, the time and frequency domains offer complementary views: in the time domain, dependencies emerge as instantaneous effects at individual time steps, while in the frequency domain, they manifest as resonances and phase shifts across specific periodic components.

Figure 1(a) shows four variables along with their spectrograms. Strong resonance patterns are evident between variables y1–y4 and y2–y3 within specific frequency bands, accompanied by clear phase delays in high frequency regions. In contrast, Figure 1(b) displays time domain correlations computed on a segment of length 100, where variables y1–y4 and y2–y3 exhibit strong similarity between steps 30 to 70, but weaker correlations elsewhere, reflecting transient dependencies. These observations suggest that combining time and frequency domain interactions provides complementary insights, capturing both transient fluctuations and periodic patterns for more comprehensive modeling of across-variable dependencies. Graph neural networks (GNNs) can represent variable interactions at each time step via adjacency matrices and have shown potential in capturing dynamic across-variable relationships. However, they incur substantial computational and memory overhead, particularly for high-dimensional datasets, since maintaining large adjacency matrices is required during both training and inference.

To overcome these limitations, we propose FACT, an efficient multi-dilated depth-wise convolution architecture tailored to capture fine-grained across-variable dependencies in both the time and frequency domains. At its core lies the DConvBlock, which draws inspiration from convolution designs in computer vision (e.g., Inception) and adapts them into lightweight depth-wise convolution variants. This design enables the extraction of meaningful across-variable interactions at each granularity without the computational burden of graph-based structures. Nevertheless, when applied to datasets with a large number of variables (e.g., Traffic, PEMS07), conventional 1D convolution typically requires stacking numerous layers to achieve a global reception field, leading to high computational costs and training instability. To address this challenge, we reconfigure the variable dimension into a two-dimensional space, allowing the use of 2D convolutions to achieve broader coverage with fewer layers. Moreover, by employing multi-dilated convolutions with progressively increasing dilation rates, we further expand the reception field while simultaneously reducing network depth and parameter count, thereby improving both computational efficiency and training stability.

Extensive experiments on twelve benchmark datasets demonstrate that FACT not only achieves state-of-the-art forecasting accuracy but also offers significant improvements in computational efficiency over attention-based models.

Our main contributions are summarized as follows:

- We propose FACT, an efficient convolution architecture that jointly leverages time- and frequency-domain variable interactions to achieve complementary modeling of multivariate time series.

- We introduce the DConvBlock, which employs depth-wise convolutions with channel-specific kernels to explicitly model variable interactions at each granularity, thereby capturing fine-grained dependencies across variables with high efficiency.

- We further reconfigure the variable dimension into a two-dimensional space and incorporate multi-dilated convolution kernels with progressively increasing dilation rates, effectively enlarging the reception field while reducing network depth.

- We conduct extensive experiments on twelve high-dimensional multivariate time series benchmarks, demonstrating that FACT consistently outperforms existing methods in forecasting accuracy while substantially lowering computational cost.

## 2 PRELIMINARIES

**Multivariate time series forecasting** Multivariate time series $\mathbf{X} \in \mathbb{R}^{C \times T}$ is a $C$-dimensional numerical sequence indexed by time, where $T$ is the number of timestamps, and $C$ is the number of variables ($C > 1$). The input time series is represented as a matrix, where each row corresponds to a variable and each column corresponds to a timestamp. The goal of multivariate time series forecasting is to predict future values of each variables based on past observations and the information from other variables. Specifically, given a historical time series $\mathbf{X}_{t-L+1:t} \in \mathbb{R}^{C \times L}$, where $L$ is the lookback length, the task is to predict the future values $\hat{\mathbf{X}}_{t+1:t+H} \in \mathbb{R}^{C \times H}$, where $H$ is the prediction length:

$$\hat{\mathbf{X}}_{t+1:t+H} = f(\mathbf{X}_{t-L+1:t}) \tag{1}$$

where $f$ is a forecasting function. The objective is to learn an accurate forecasting function $f$ that minimizes the prediction error between the predicted time series $\hat{\mathbf{X}}_{t+1:t+H}$ and the ground truth $\mathbf{Y}_{t+1:t+H} \in \mathbb{R}^{C \times H}$. The prediction error can be quantified using the Mean Squared Error (MSE) loss function: $\mathcal{L} = \left\| \mathbf{Y}_{t+1:t+H} - \hat{\mathbf{X}}_{t+1:t+H} \right\|_2^2$. Our focus is on multivariate time series forecasting, and it is beyond the scope of this work to assume or utilize any spatial ordering among variables.

**Discrete Fourier transform** Given a discrete time series $x[n] \in \mathbb{R}^N$, its frequency representation can be obtained via the Discrete Fourier Transform (DFT), which decomposes the signal into a sum of sinusoids at different frequencies. The DFT is defined as:

$$\mathcal{F}[k] = \sum_{n=0}^{N-1} x[n] \cdot e^{-j\frac{2\pi}{N}kn}, \quad k = 0, 1, \ldots, N-1 \tag{2}$$

Each frequency component $\mathcal{F}[k]$ is a complex number and can be expressed in terms of its real and imaginary parts as:

$$\mathcal{F}[k] = \mathcal{R}(\mathcal{F}[k]) + j \cdot \mathcal{I}(\mathcal{F}[k]) = \mathcal{R}_k + j\mathcal{I}_k \tag{3}$$

where the real part $\mathcal{R}_k$ and imaginary part $\mathcal{I}_k$ are computed as:

$$\mathcal{R}_k = \sum_{n=0}^{N-1} x[n] \cdot \cos\left(\frac{2\pi}{N}kn\right), \mathcal{I}_k = -\sum_{n=0}^{N-1} x[n] \cdot \sin\left(\frac{2\pi}{N}kn\right) \tag{4}$$

From these components, we can compute the amplitude $|\mathcal{F}[k]|$ and phase $\phi[k]$ of each frequency $k$ as follows:

$$|\mathcal{F}[k]| = \sqrt{\mathcal{R}_k^2 + \mathcal{I}_k^2}, \phi[k] = \arctan(\frac{\mathcal{I}_k}{\mathcal{R}_k}) \tag{5}$$

Given the magnitude $|\mathcal{F}[k]|$ and phase $\phi[k]$, the real and imaginary parts can be reconstructed as:

$$\mathcal{R}_k = |\mathcal{F}[k]| \cdot \cos(\phi[k]), \mathcal{I}_k = |\mathcal{F}[k]| \cdot \sin(\phi[k]) \tag{6}$$

Thus, the representation of a frequency component can be equivalently expressed in either Cartesian form $(\mathcal{R}_k, \mathcal{I}_k)$ or polar form $(|\mathcal{F}[k]|, \phi[k])$. Since cross-variable frequency correlations are often manifested through energy distributions and phase relationships, we adopt the amplitude–phase representation of the frequency spectrum instead of its Cartesian decomposition. This choice allows for a more direct and interpretable characterization of inter-variable dependencies in the frequency domain. Experimental results in Appendix D further validate the superiority of this representation over alternative frequency modeling approaches.

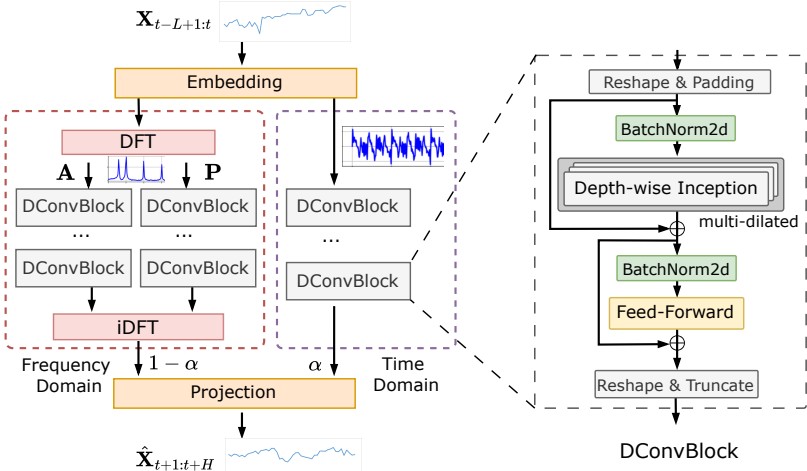

Figure 2: The framework of FACT. The time series is projected into both the time and frequency domains, where depth-wise DConvBlocks are employed to capture fine-grained variable interactions. Each DConvBlock adopts a multi-dilated convolution structure to enlarge the reception field while reducing computational complexity.

## 3 METHODOLOGY

The overall framework of FACT is illustrated in Figure 2. At its core, FACT employs the proposed CNN-based DConvBlock to capture inter-variable dependencies from both the time and frequency domains. The input time series is first projected into a high-dimensional embedding space $\mathbf{X}_e \in \mathbb{R}^{C \times D}$, where $D$ denotes the hidden dimension, to enhance representational capacity. We also compared modeling in the raw time-series space and the embedding space in Appendix E.1.

To achieve comprehensive modeling of dependencies, we adopt a dual-domain strategy. In the time domain, DConvBlocks are directly applied to the embeddings to capture instantaneous interactions among variables. In the frequency domain, the embeddings are transformed via the Fourier transform, and DConvBlocks are then employed to model dependencies across amplitude and phase components. Since Parseval's theorem guarantees energy equivalence between the time and frequency domains, and instantaneous interactions (time domain) and periodic interactions (frequency domain) provide complementary perspectives, we introduce a weighting factor $\alpha$ to balance temporal and spectral information, thereby enabling a unified characterization of across-variable dependencies. The effects of different $\alpha$ can be found in Appendix B.3.

Technically, in each domain, the DConvBlock first reconfigures the one-dimensional variables into a two-dimensional form, reducing the effective distance between variables. It then applies multi-dilated 2D depth-wise convolutions with progressively increasing dilation rates to capture fine-grained dependencies while substantially enlarging the reception field. A subsequent feed-forward network further enhances the learned variable representations. Structurally, the DConvBlock resembles a Transformer block, but replaces the attention mechanism with the proposed multi-dilated depth-wise convolutions. This substitution yields a simpler yet more efficient design that retains strong representational power while greatly reducing training cost compared to attention.

### 3.1 TIME AND FREQUENCY MODELING

**Time domain** Multivariate time series often exhibit dynamic and time-varying instantaneous interactions among all variables. To effectively capture these dependencies, we employ $N$ stacked DConvBlocks. Given the input embedding $\mathbf{X}_e$, the output of time-domain modeling $\mathbf{X}_{l+1}^t \in \mathbb{R}^{C \times D}$ can be computed as:

$$\mathbf{X}_0^t = \mathbf{X}_e, \tag{7}$$

$$\mathbf{X}_{l+1}^t = \text{DConvBlock}(\mathbf{X}_l^t), l \in \{0, 1, , , , N-1\}, \tag{8}$$

**Frequency domain** Multivariate time series also show dependencies across variables at different frequencies, such as low-frequency trends and high-frequency fluctuations. To capture this, we model fine-grained variable interactions in the frequency domain. Specifically, given the input embedding $\mathbf{X}_e$, we apply the Discrete Fourier Transform (DFT) to project it into the frequency domain:

$$\mathcal{F} = \text{DFT}(\mathbf{X}_e) = \mathcal{R}(\mathcal{F}) + j \cdot \mathcal{I}(\mathcal{F}) \in \mathbb{R}^{C \times \lceil \frac{(D+1)}{2} \rceil}, \tag{9}$$

where $\mathcal{R}(\mathcal{F})$ and $\mathcal{I}(\mathcal{F})$ denote the real part and imaginary parts, respectively. Then we compute the amplitude and phase of each frequency:

$$\mathbf{A} = |\mathcal{F}[k]| = \sqrt{\mathcal{R}_k^2 + \mathcal{I}_k^2}, \mathbf{P} = \phi[k] = \arctan(\frac{\mathcal{I}_k}{\mathcal{R}_k}), \tag{10}$$

The amplitude $\mathbf{A}$ of each frequency component captures the energy or strength of oscillations at specific frequencies, reflecting how strongly variables co-vary in terms of amplitude. The phase $\mathbf{P}$ encodes the relative timing of these oscillations, indicating temporal lead-lag relationships or synchronization patterns between variables.

For simplicity, we separately model the amplitude and phase components using $N$ DConvBlocks, which aligns with the physical interpretation of Fourier components and enhances its expressiveness for complex multivariate temporal dependencies. After $N$ stacked DConvBlocks to reconstruct the amplitude and phase, we get the output of frequency domain and use inverse DFT to transform it into the time domain to get $\mathbf{X}_{l+1}^f \in \mathbb{R}^{C \times D}$. The overall process is summarized as follows:

$$\mathbf{A}_0 = \mathbf{A}, \mathbf{P}_0 = \mathbf{P}, \tag{11}$$

$$\mathbf{A}_{l+1} = \text{DConvBlock}(\mathbf{A}_l), l = 0, 1, ..., N-1, \tag{12}$$

$$\mathbf{P}_{l+1} = \text{DConvBlock}(\mathbf{P}_l), l = 0, 1, ..., N-1, \tag{13}$$

$$\mathbf{X}_{l+1}^f = \text{iDFT}(\mathbf{A}_{l+1} \cdot \cos(\mathbf{P}_{l+1}), j \cdot (\mathbf{A}_{l+1} \cdot \sin(\mathbf{P}_{l+1}))), \tag{14}$$

Finally, we fuse the outputs from the time and frequency domains using a weighting factor $\alpha$ to obtain the final representation: $\mathbf{X}_{l+1} = \alpha \cdot \mathbf{X}_{l+1}^t + (1-\alpha) \cdot \mathbf{X}_{l+1}^f$. The parameter $\alpha$ enables flexible control over the relative contributions of each domain. Subsequently, a linear transformation along the hidden dimension is applied to project $\mathbf{X}_{l+1}$ into the predicted sequence $\hat{\mathbf{X}}_{t+1:t+H}$.

## 3.2 DCONVBLOCK

The DConvBlock is designed to capture the dynamic interactions among different variables at each granularity, and it mainly consists of two sub-layers: a multi-dilated depth-wise convolution structure and a fully connected feed-forward network. Batch normalization and residual connections are applied within each sub-layer. Concretely, we first transform the 1D variable representation into 2D space and then select 2D convolution to accelerate the training efficiency. Then the multi-dilated depth-wise convolution structure is employed to capture fine-grained inter-variable interactions while significantly expanding the reception field. Subsequently, the feed-forward network is used to enhance the variable representations and the 2D variable is reshaped back into a 1D space to restore the original data format.

**Depth-wise structure** Convolution structures effectively capture local dependencies among variables, but traditional convolutions compute the filters across all channels, limiting their ability to model interactions specific to each granularity. To address this limitation, we adopt a depth-wise convolution design, assigning each input channel a separate convolution kernel. By allowing each kernel to operate independently on a specific granularity, the depth-wise convolution enables granularity-specific learning of variable interactions.

Meanwhile, it is worth noting that several popular convolution architectures, such as standard 2D convolution, Inception (Szegedy et al., 2015), and ResNet (He et al., 2016), can serve as the convolution backbone. In this work, we adopt the Inception structure and modify it into a depth-wise Inception variant due to its stronger representational capacity compared to standard 2D convolution. At the same time, our experiments in Appendix C demonstrate that even standard depth-wise 2D convolutions achieve competitive performance, though slightly inferior to Inception overall. This observation further confirms that the effectiveness of our approach arises not from the specific choice of Inception, but from the underlying architectural design itself.

**2D modeling**  With the same number of convolution kernel parameters and an equal number of model layers, 2D convolution can achieve a larger reception field compared to 1D convolution. Assume that the length of a 1D tensor is $S$, which we reshape into a 2D tensor of size approximately $\sqrt{S} \times \sqrt{S}$. If the kernel size of the 1D convolution is $s$, then the kernel size of 2D convolution with the same number of parameters is approximately $\sqrt{s} \times \sqrt{s}$. For simplicity, we ignore stride and padding effects in the following analysis.

For 1D convolution, the number of layers $l_{1D}$ required to cover the entire length $S$ is:

$$l_{1D} = \lceil \frac{S-1}{s-1} + 1 \rceil \tag{15}$$

Similarly, for 2D convolution, the number of layers $l_{2D}$ to cover the entire $\sqrt{S} \times \sqrt{S}$ input is:

$$l_{2D} = \lceil \frac{\sqrt{S}-1}{\sqrt{s}-1} + 1 \rceil \tag{16}$$

Then the percentage reduction in the number of layers when using 2D convolution compared to 1D convolution is given by (if the small effect of $\lceil \cdot \rceil$ on the results is ignored and we set $S - 1 \approx S, \sqrt{S} - 1 \approx \sqrt{S}, s - 1 \approx s, \sqrt{s} - 1 \approx \sqrt{s}$):

$$r = \frac{l_{1D} - l_{2D}}{l_{2D}} \approx \frac{\sqrt{S}}{\sqrt{s}} - 2 + \frac{2}{\frac{\sqrt{S}}{\sqrt{s}} + 1} \tag{17}$$

From this approximate relation, it is evident that as the length $S$ increases, the relative reduction of the layers required for the 2D convolution also increases. Based on this observation, we convert the variables into 2D space to leverage 2D convolution architectures for modeling dynamic inter-variable interactions efficiently. We focus on multivariate time series forecasting, and do not assume or leverage any inherent spatial order among the variables. We reshape the input sequence into a 2D format by row-major order. Specifically, we pad and convert $\mathbf{X}_l^t \in \mathbb{R}^{C \times D}, \mathbf{A}_l, \mathbf{P}_l \in \mathbb{R}^{C \times \lceil \frac{(D+1)}{2} \rceil}$ to $\mathbf{X}_{l,2D}^t \in \mathbb{R}^{H \times W \times D}, \mathbf{A}_{l,2D}, \mathbf{P}_{l,2D} \in \mathbb{R}^{H \times W \times \lceil \frac{(D+1)}{2} \rceil}$ at the beginning of DConvBlock, where $H = W = \lceil \sqrt{C} \rceil$. Experiments in Appendix C validate the effectiveness of 2D modeling over 1D.

**Multi-dilated architecture**  To further enhance the ability to capture variable dependencies without increasing parameters count or stacking numerous layers, we incorporate dilated convolutions into the architecture. As shown in Figure 3, by applying multiple dilation rates, the reception field can be significantly expanded, allowing the model to efficiently capture variable interactions while ensuring a more dense and flexible coverage of the input space at each scale. This design complements the depth-wise structure to jointly achieve fine-grained modeling and efficient reception field expansion. Experiments in Appendix C validate the effectiveness of multi-dilated design.

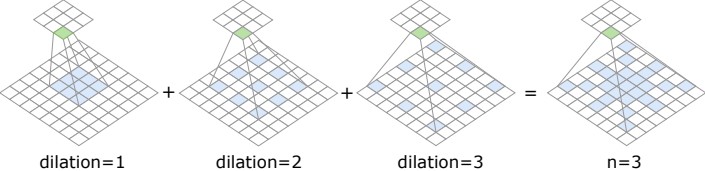

Figure 3: Multi-dilated architecture. $n = 3$ denotes 3 increasing dilation sizes.

In summary, let MDInception represent the multi-dilated depth-wise Inception module, BN represent batch normalization and use $n$ distinct increasing dilation rates from 1. Then for $\mathbf{X}_{l,2D}^t, \mathbf{A}_{l,2D}, \mathbf{P}_{l,2D}$, the corresponding outputs, denoted as $\mathbf{X}_{l,c}^t \in \mathbb{R}^{H \times W \times D}, \mathbf{A}_{l,c}, \mathbf{P}_{l,c} \in$

$\mathbb{R}^{H \times W \times \lceil \frac{(D+1)}{2} \rceil}$, can be computed as:

$$\mathbf{X}_{l,c}^t = \mathbf{X}_{l,2D}^t + \sum_{i=1}^{n} \text{MDInception}(\text{BN}(\mathbf{X}_{l,2D}^t))_{\text{dilation=i}}, \tag{18}$$

$$\mathbf{A}_{l,c} = \mathbf{A}_{l,2D} + \sum_{i=1}^{n} \text{MDInception}(\text{BN}(\mathbf{A}_{l,2D}))_{\text{dilation=i}}, \tag{19}$$

$$\mathbf{P}_{l,c} = \mathbf{P}_{l,2D} + \sum_{i=1}^{n} \text{MDInception}(\text{BN}(\mathbf{P}_{l,2D}))_{\text{dilation=i}} \tag{20}$$

**Feed-Forward Networks**  After modeling the fine-grained across-variable interactions through the multi-dilated depth-wise convolution structure, we employ a fully connected feed-forward network (FFN) among the hidden dimension to further integrate the information. FFN acts as a feature transformation module to enhance the variable representations and produce the output $\mathbf{X}_{l,f}^t, \mathbf{A}_{l,f}, \mathbf{P}_{l,f}$. This process can be summarized as follows:

$$\mathbf{X}_{l,F}^t = \mathbf{X}_{l,c}^t + \text{FFN}(\text{BN}(\mathbf{X}_{l,c}^t)), \tag{21}$$

$$\mathbf{A}_{l,F} = \mathbf{A}_{l,c} + \text{FFN}(\text{BN}(\mathbf{A}_{l,c})), \tag{22}$$

$$\mathbf{P}_{l,F} = \mathbf{P}_{l,c} + \text{FFN}(\text{BN}(\mathbf{P}_{l,c})). \tag{23}$$

Finally, $\mathbf{X}_{l,F}^t, \mathbf{A}_{l,F}, \mathbf{P}_{l,F}$ are reshaped from 2D space back into the original 1D space. During this process, we also truncate it along the variable dimension (if necessary) to ensure that its shape matches the input dimension, resulting in the output of $l$-layers DConvBlocks $\mathbf{X}_{l+1}^t, \mathbf{A}_{l+1}, \mathbf{P}_{l+1}$.

## 4 EXPERIMENTS

### 4.1 SETUP

**Datasets.** We evaluate the proposed FACT on twelve datasets, encompassing diverse variable relationships. For long-term forecasting (prediction lengths: $\{96, 192, 336, 720\}$), we use three high-variable real-world datasets: ECL, Traffic, and Solar-Energy (from LSTNet (Lai et al., 2018)). For short-term forecasting (prediction lengths: $\{12, 24, 48, 96\}$), we include nine public datasets: PEMS03, PEMS04, PEMS07, PEMS08 (from SCINet (Liu et al., 2022)), PEMSD7 (Sen et al., 2019), PEMS-BAY, METR-LA (Li et al., 2018), electricity and Traffic2. The lookback length of all datasets are fixed to 96 for fair comparison. Dataset details are in Appendix B.1.

**Baselines.** We include twelve latest models as baselines: one variable dependent models: Times-Net (Wu et al., 2023a); two variable attention models: iTransformer (Liu et al., 2024), TQNet (Lin et al., 2025) and TimeXer (Wang et al., 2024b); one variable CNN model: ModernTCN (Luo & Wang, 2024); one variable Mamba model: TimePro (Ma et al., 2025); one variable clustering model: DUET (Qiu et al., 2025); two variable MLP models: TSMixer (Ekambaram et al., 2023) and SOFTS (Han et al., 2024); two variable independent models: PatchTST (Nie et al., 2023) and CycleNet (Lin et al., 2024a); and one Graph-based model: TimeFilter (Hu et al., 2025). The hyperparameter configuration is provided in Appendix B.2 and the hyperparameter sensitivity analysis is provided in Appendix B.3.

### 4.2 MAIN RESULTS

Table 1 and Table 2 summarize the overall performance of FACT and baseline models across all datasets, evaluated using Mean Squared Error (MSE) and Mean Absolute Error (MAE). The results clearly demonstrate that FACT consistently achieves state-of-the-art performance on nearly all datasets, highlighting its strong ability to model fine-grained inter-variable interactions effectively. Specifically, on the PEMS03 dataset (Table 1), our FACT achieves remarkable improvements over existing baselines: it reduces the MSE by 23.9% compared to the variable-attention model iTransformer, by 22.5% compared to the variable-CNN model ModernTCN, by 20.4% compared to the graph-based model TimeFilter and by 27.1% compared to the variable-independent model CycleNet. And on METR-LA dataset (Table 2), our FACT reduces the MSE by 15.8% compared

Table 1: Full results on part of datasets. The input length $L$ is fixed as 96 and the results are averaged from all prediction horizons. The results of other models are sourced from iTransformer (Liu et al., 2024). The best results are highlighted in **bold**. $1^{st}$ denote the count of the best performance.

| Models | PEMS03 | | PEMS04 | | PEMS07 | | PEMS08 | | PEMSD7 | | PEMS-BAY | | $1^{st}$ |
|---|---|---|---|---|---|---|---|---|---|---|---|---|---|
| | MSE | MAE | MSE | MAE | MSE | MAE | MSE | MAE | MSE | MAE | MSE | MAE | |
| TimesNet(2023a) | 0.173 | 0.264 | 0.148 | 0.263 | 0.133 | 0.241 | 0.214 | 0.293 | 0.569 | 0.436 | 0.669 | 0.375 | 0 |
| PatchTST(2023) | 0.150 | 0.260 | 0.171 | 0.283 | 0.165 | 0.267 | 0.242 | 0.285 | 0.485 | 0.412 | 0.596 | 0.382 | 0 |
| CycleNet(2024a) | 0.118 | 0.225 | 0.119 | 0.231 | 0.113 | 0.189 | 0.150 | 0.228 | 0.503 | 0.429 | 0.604 | 0.397 | 0 |
| SOFTS(2024) | 0.104 | 0.210 | 0.102 | 0.208 | 0.087 | 0.184 | 0.138 | 0.219 | 0.503 | 0.429 | 0.604 | 0.397 | 0 |
| TSMixer(2023) | 0.158 | 0.263 | 0.129 | 0.247 | 0.117 | 0.221 | 0.201 | 0.294 | 0.488 | 0.394 | 0.523 | 0.336 | 0 |
| ModernTCN(2024) | 0.111 | 0.226 | 0.103 | 0.217 | 0.118 | 0.208 | 0.230 | 0.268 | 0.374 | 0.370 | 0.436 | 0.330 | 0 |
| TimePro(2025) | 0.119 | 0.225 | 0.114 | 0.226 | 0.096 | 0.200 | 0.252 | 0.259 | 0.412 | 0.370 | 0.577 | 0.389 | 0 |
| DUET(2025) | 0.110 | 0.219 | 0.113 | 0.227 | 0.091 | 0.198 | 0.130 | 0.229 | 0.360 | 0.336 | 0.447 | 0.326 | 0 |
| TQNet(2025) | 0.097 | 0.203 | 0.091 | 0.197 | 0.075 | 0.171 | 0.142 | 0.229 | 0.361 | 0.343 | 0.427 | 0.315 | 0 |
| iTransformer(2024) | 0.113 | 0.221 | 0.119 | 0.231 | 0.113 | 0.189 | 0.150 | 0.228 | 0.382 | 0.354 | 0.447 | 0.319 | 0 |
| TimeXer(2024b) | 0.111 | 0.223 | 0.110 | 0.227 | 0.093 | 0.199 | 0.233 | 0.269 | 0.337 | 0.319 | 0.394 | 0.288 | 0 |
| TimeFilter(2025) | 0.108 | 0.217 | 0.114 | 0.226 | 0.101 | 0.207 | 0.139 | 0.230 | 0.498 | 0.404 | 0.547 | 0.358 | 0 |
| **FACT** | **0.086** | **0.194** | **0.087** | **0.194** | **0.073** | **0.170** | **0.107** | **0.200** | **0.330** | **0.316** | **0.372** | **0.267** | **12** |

Table 2: Full results on other datasets.

| Models | ECL | | Traffic | | Solar-Energy | | electricity | | Traffic2 | | METR-LA | | $1^{st}$ |
|---|---|---|---|---|---|---|---|---|---|---|---|---|---|
| | MSE | MAE | MSE | MAE | MSE | MAE | MSE | MAE | MSE | MAE | MSE | MAE | |
| TimesNet(2023a) | 0.192 | 0.295 | 0.620 | 0.336 | 0.301 | 0.319 | 965.69 | 2.265 | 0.272 | 0.258 | 0.908 | 0.520 | 0 |
| PatchTST(2023) | 0.189 | 0.276 | 0.454 | 0.286 | 0.236 | 0.266 | 410.90 | 1.648 | 0.234 | 0.204 | 0.762 | 0.488 | 0 |
| CycleNet(2024a) | 0.168 | 0.259 | 0.472 | 0.301 | 0.210 | 0.261 | 178.53 | 0.988 | 0.292 | 0.254 | 0.748 | 0.503 | 0 |
| SOFTS(2024) | 0.174 | 0.264 | 0.409 | **0.267** | 0.229 | 0.256 | 264.81 | 1.304 | 0.224 | 0.192 | 0.748 | 0.503 | 1 |
| TSMixer(2025) | 0.186 | 0.287 | 0.522 | 0.357 | 0.260 | 0.297 | 257.08 | 1.222 | 0.244 | 0.232 | 0.833 | 0.490 | 0 |
| ModernTCN(2024) | 0.197 | 0.282 | 0.546 | 0.348 | 0.211 | 0.310 | 509.11 | 1.899 | 0.252 | 0.245 | 0.808 | 0.500 | 0 |
| TimePro(2025) | 0.169 | 0.262 | 0.443 | 0.287 | 0.232 | 0.266 | 200.68 | 1.093 | 0.244 | 0.218 | 0.787 | 0.533 | 0 |
| DUET(2025) | 0.170 | 0.265 | 0.469 | 0.289 | 0.220 | 0.263 | 244.13 | 1.244 | 0.279 | 0.244 | 0.813 | 0.487 | 0 |
| TQNet(2025) | 0.164 | 0.259 | 0.445 | 0.276 | 0.198 | 0.256 | 203.41 | 1.038 | 0.234 | 0.200 | 0.749 | 0.489 | 0 |
| iTransformer(2024) | 0.178 | 0.270 | 0.428 | 0.282 | 0.233 | 0.262 | 384.22 | 1.656 | 0.221 | 0.193 | 0.837 | 0.536 | 0 |
| TimeXer(2024b) | 0.171 | 0.270 | 0.466 | 0.287 | 0.202 | 0.269 | 261.84 | 1.310 | 0.217 | 0.195 | 0.750 | 0.496 | 0 |
| TimeFilter(2025) | 0.158 | 0.256 | **0.407** | 0.268 | 0.223 | **0.250** | 193.29 | 1.054 | 0.227 | 0.199 | 0.819 | 0.493 | 2 |
| **FACT** | **0.157** | **0.254** | 0.425 | 0.276 | **0.196** | 0.254 | **159.73** | **0.922** | **0.204** | **0.179** | **0.704** | **0.456** | **9** |

to iTransformer, by 12.8% compared to ModernTCN, by 14.0% compared to TimeFilter and by 5.3% compared to CycleNet. These results demonstrate the robustness and effectiveness of our proposed architecture in capturing dynamic, fine-grained inter-variable interactions. Experiments under longer lookback lengths in Appendix E.7 demonstrate that FACT can effectively leverage historical information. Furthermore, the performance promotion of our DConvBlock on other baselines, presented in Appendix E.2, further indicates that FACT enhances variable representations. Prediction showcases are provided in Appendix G.2, illustrating the accuracy of prediction across diverse scenarios.

## 5 COMPARISON ON THE TARGET FORECASTING HORIZON

Although averaging errors across all forecasting horizons is the standard evaluation protocol in multivariate time series forecasting, as adopted by recent representative works such as PatchTST, iTransformer, and TimeFilter, evaluation at specific target horizon $H$ (single time step) is also important.

Therefore, we report performance with several different target horizons (target time step $t+H$): t+3, t+6, t+12, and t+24. For a comprehensive comparison, we include several recent and competitive baselines: the latest attention-based models (iTransformer, TimeXer), the latest graph-based model (TimeFilter), and the latest Mamba-based model (TimePro). The results are summarized in Table 3, using MSE as the evaluation metric. Across all baselines and target horizons, FACT consistently demonstrates superior performance. Notably, as the forecasting horizon increases, the performance

gap between FACT and other baselines generally widens. These results further validate the effectiveness of our proposed method under various evaluation settings.

Table 3: Performance comparison on the target forecasting horizon.

| $H$ | Model | PEMS03 | PEMS04 | PEMS08 | METR-LA | Traffic2 | PEMSD7 | PEMS-BAY |
|---|---|---|---|---|---|---|---|---|
| 3 | **FACT** | **0.049** | **0.061** | **0.058** | **0.273** | **0.168** | **0.153** | **0.162** |
| | TimeFilter | 0.050 | 0.064 | 0.058 | 0.277 | 0.187 | 0.163 | 0.210 |
| | TimeXer | 0.062 | 0.070 | 0.167 | 0.283 | 0.181 | 0.156 | 0.172 |
| | TimePro | 0.058 | 0.069 | 0.161 | 0.313 | 0.183 | 0.163 | 0.218 |
| | iTransformer | 0.063 | 0.069 | 0.161 | 0.326 | 0.188 | 0.163 | 0.175 |
| 6 | **FACT** | **0.059** | **0.068** | **0.069** | **0.438** | **0.184** | **0.249** | **0.270** |
| | TimeFilter | 0.063 | 0.075 | 0.070 | 0.443 | 0.206 | 0.284 | 0.350 |
| | TimeXer | 0.073 | 0.078 | 0.186 | 0.439 | 0.196 | 0.253 | 0.276 |
| | TimePro | 0.071 | 0.079 | 0.188 | 0.465 | 0.200 | 0.265 | 0.332 |
| | iTransformer | 0.075 | 0.081 | 0.179 | 0.468 | 0.206 | 0.269 | 0.290 |
| 12 | **FACT** | **0.073** | **0.077** | **0.085** | **0.668** | **0.203** | **0.336** | **0.371** |
| | TimeFilter | 0.082 | 0.091 | 0.092 | 0.692 | 0.222 | 0.441 | 0.547 |
| | TimeXer | 0.089 | 0.090 | 0.214 | 0.670 | 0.211 | 0.349 | 0.376 |
| | TimePro | 0.092 | 0.097 | 0.240 | 0.686 | 0.218 | 0.375 | 0.455 |
| | iTransformer | 0.098 | 0.102 | 0.218 | 0.677 | 0.221 | 0.387 | 0.413 |
| 24 | **FACT** | **0.098** | **0.097** | **0.117** | **0.927** | **0.222** | **0.414** | **0.451** |
| | TimeFilter | 0.129 | 0.131 | 0.146 | 1.014 | 0.232 | 0.661 | 0.875 |
| | TimeXer | 0.124 | 0.115 | 0.268 | 0.943 | 0.227 | 0.427 | 0.478 |
| | TimePro | 0.132 | 0.132 | 0.352 | 0.956 | 0.229 | 0.482 | 0.609 |
| | iTransformer | 0.143 | 0.143 | 0.295 | 0.962 | 0.233 | 0.505 | 0.546 |

## 6 COMPARED WITH ATTENTION MECHANISM

Since our DConvBlock adopts a Transformer-like architecture, its key distinction lies in replacing the attention mechanism with multi-dilated depth-wise convolutions to capture fine-grained variable interactions. To evaluate the efficacy and efficiency of this design, we conduct a detailed comparison between multi-dilated depth-wise convolutions and attention mechanisms in terms of forecasting accuracy, training time, and memory cost. As shown in Figure 4, we evaluate on eight datasets with varying numbers of variables under identical hyperparameter settings to ensure fairness. The results show that our method not only achieves superior forecasting performance, which is consistent with the dynamic changes of inter-variable interactions, but also delivers substantial efficiency gains over attention-based approaches. In particular, on large-scale datasets such as Traffic (862 variables), PEMS07 (883 variables), and Traffic2 (963 variables), our approach reduces the computational cost by up to 50%. These advantages stem from the multi-dilated depth-wise convolution design, whose complexity primarily depends on kernel size and channel dimension, rather than input sequence length as in attention. More comparisons with TimeFilter and ModernTCN are provided in Appendix E.5, further demonstrating the superior efficiency of our FACT over existing baselines.

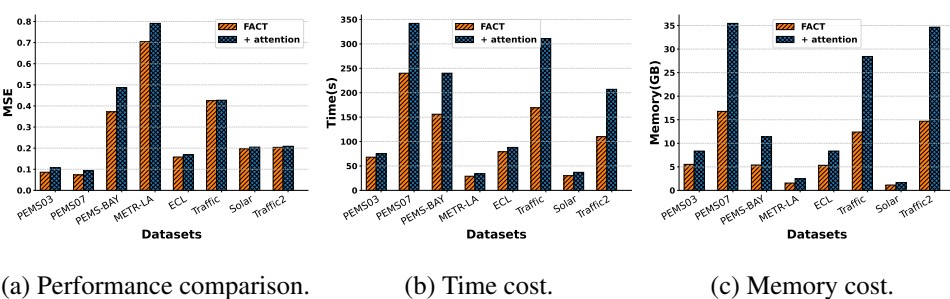

(a) Performance comparison.    (b) Time cost.    (c) Memory cost.

Figure 4: Compared with attention mechanism.

# 7 ABLATION ANALYSIS

**Modules ablation**  As illustrated in Figures 5, we perform an ablation study on two key components of FACT: the multi-dilated depth-wise Inception module and the feed-forward network (FFN). We denote the variant without the Inception module as w/o MDInception and the variant without the FFN as w/o FFN. The multi-dilated depth-wise Inception module is responsible for capturing variable interactions at each granularity, while the FFN integrates information across granularities to enhance variable representations. The results show that both components are essential, each contributing complementary benefits to modeling different aspects of the data. Notably, removing the Inception module leads to a more substantial performance drop in most cases, highlighting the critical role of explicitly modeling inter-variable interactions.

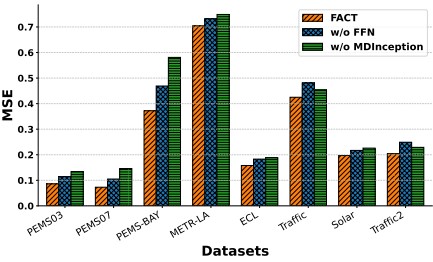

Figure 5: Modules ablation.

**Time and frequency modeling ablation**  As shown in Table 4, we further analyze the necessity of modeling variable interactions from both the time and frequency domains. We denote the variant without frequency-domain modeling as w/o Frequency and the variant without time-domain modeling as w/o Time. The results demonstrate that the two domains provide complementary information. Time-domain modeling captures transient and instantaneous variable interactions, while frequency-domain modeling reveals periodic variable dependencies. Combining both perspectives enables a more comprehensive characterization of across-variable interactions. Meanwhile, the effects of the domain weight factor $\alpha$ can be found in Appendix B.3.

Table 4: Ablation results of time domain and frequency domain. We report the average results.

| Models | PEMS03 | | PEMS04 | | METR-LA | | ECL | | Traffic | | Solar | | Traffic2 | |
|---|---|---|---|---|---|---|---|---|---|---|---|---|---|---|
| | MSE | MAE | MSE | MAE | MSE | MAE | MSE | MAE | MSE | MAE | MSE | MAE | MSE | MAE |
| **FACT** | **0.086** | **0.194** | **0.087** | **0.194** | **0.704** | **0.456** | **0.157** | **0.254** | **0.425** | **0.276** | **0.197** | **0.259** | **0.204** | **0.179** |
| **w/o Time** | 0.089 | 0.197 | 0.089 | 0.198 | 0.755 | 0.483 | 0.162 | 0.258 | 0.436 | 0.286 | 0.202 | 0.263 | 0.215 | 0.188 |
| **w/o Frequency** | 0.087 | 0.196 | 0.090 | 0.201 | 0.722 | 0.459 | 0.162 | 0.258 | 0.426 | 0.276 | 0.198 | 0.256 | 0.212 | 0.180 |

# 8 CONCLUSION

In this paper, we propose FACT, a novel convolution architecture that efficiently captures fine-grained variable interactions in both the time and frequency domains. To this end, we design a specialized DConvBlock, which leverages a depth-wise Inception module to dynamically extract fine-grained cross-variable interactions. Additionally, we reconfigure the original 1D variables into a 2D form to reduce the required model layers and further employs a multi-dilated convolution design with progressively increasing dilation rates, substantially enlarging the reception field without increasing the number of parameters. Extensive experiments reveal three key findings: (1) modeling interactions in both time and frequency domains is crucial for comprehensive characterization of across-variable dependencies; (2) fine-grained modeling significantly enhances the ability to capture complex variable relationships; and (3) the proposed DConvBlock achieves superior efficiency and effectiveness compared with conventional attention mechanisms.

ACKNOWLEDGMENTS

This work is supported by grants from the Research Grants Council of Hong Kong Special Administrative Region, China (No. PolyU 15205224), and Smart Cities Research Institute (SCRI) P0051036-P0050643.

REPRODUCIBILITY STATEMENT

To ensure reproducibility, we provide the implementation details in Appendix B and release the code at the github repository: `https://github.com/wanghq21/FACT`.

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

# A  RELATED WORK

Exploring variable relationships in multivariate time series forecasting has garnered significant attention, with existing methods categorized into Variable Dependent, Variable Independent, Attention-based, Cluster-based, MLP-based, CNN-based, Mamba-based, and Graph-based approaches.

**Variable Dependent method.** Early works embed all variables into a hidden vector and focus on capturing the time dependency within the time series, such as cyclical characteristics, trend changes, etc. This method could increase the model's representational capability but lack the modeling of variable relationship and have poor generalization, such as Informer (Zhou et al., 2021), Autoformer (Wu et al., 2021), FEDformer (Zhou et al., 2022), MICN (Wang et al., 2023), Times-Net (Wu et al., 2023a), Pathformer (Chen et al., 2024b), TimeMixer (Wang et al., 2024a) and TimeMixer++ (Wang et al., 2025b). TimeMixer++ serves as the latest model for capturing the temporal dependencies such as cycles and trends within time series through sophisticated designs that require great training time and memory costs.

**Variable Independent method.** Variable Independent method treats the multivariate variables as multiple independent individual variables, including PatchTST (Nie et al., 2023), DLinear (Zeng et al., 2023), SegRNN (Lin et al., 2023), SparseTSF (Lin et al., 2024b) and CycleNet (Lin et al., 2024a), which is more generalizable, but the representation capability is reduced and the complex relationship between variables are ignored. The difference between a multivariate time series and a univariate time series is that a multivariate time series contains complex variable relationships, such as correlation, cointegration, causation, etc. Therefore, the modeling of multivariate time series should be different from the modeling of univariate time series and include how to deal with the complex variables relationship.

**Attention-based method.** Attention mechanism excels in in capturing correlations between different elements, so many works use it to model variable relationship: TQNet (Lin et al., 2025), iTransformer (Liu et al., 2024), Leddam (Yu et al., 2024), CARD (Xue et al., 2024), Crossformer (Zhang & Yan, 2023), Fredformer (Piao et al., 2024) and TimeXer (Wang et al., 2024b). However, for multivariate time series with a relatively large variable number, attention mechanism will require huge time and memory costs.

**Cluster-based method.** Clustering methods enhance the modeling of variable relationship by clustering variables with strong correlations into the same class (Qiu et al., 2025; Chen et al., 2024a). But the process of finding variable similarity in clustering algorithms requires the same quadratic complexity as the attention mechanism, which can be costly.

**MLP-based method.** MLP-based method directly uses MLP to model correlations between variables, including TSMixer (Ekambaram et al., 2023) and SOFTS (Han et al., 2024). This method is lighter, but not sufficient to cope with the complex variable relationship.

**CNN-based method.** CNN-based method uses CNN to model the variable correlation, including ModernTCN (Luo & Wang, 2024), which fuses the modeling of variable correlation into the grouping operation of convolution but requires a high number of parameters and training cost.

**Mamba-based method.** Mamba-based method transfers Mamba to the time series forecasting to delegates the extraction of inter-variate correlations, including TimePro (Ma et al., 2025), SMamba (Wang et al., 2025c).

**Graph-based method.** Graph-based method uses graph structure to model variable correlations, where each node represents each variable and the edges represent correlations between variables, including MTGNN (Wu et al., 2020) and TimeFilter (Hu et al., 2025). Graph can effectively capture the fine-grained variable relationship at each time step, but how to construct and efficiently store the adjacency matrix is a challenge. Moreover, graph structure is not very efficient in handling long sequences and usually requires great memory cost.

Unlike the above approaches that explore variable relationships in terms of the whole variable, this paper focuses on highly efficient fine-grained variable dependencies. We capture the variable interactions from both the time and frequency domains and then utilize multi-dilated depth-wise convolution structure to capture the different fine-grained inter-variable dependencies at a much lower training time and memory cost.

## B IMPLEMENTATION DETAILS

### B.1 DATASET DETAILS

The details of the experimental datasets are summarized as follows: (1) *ECL* (Wu et al., 2021): Records hourly electricity consumption of 321 customers from 2012 to 2014. (2) *electricity* (Salinas et al., 2020): The electricity dataset is a hourly time series of electricity consumption of 370 customers. (3) *Traffic* (Wu et al., 2021): Hourly road occupancy rates from sensors on San Francisco Bay area freeways, collected by the California Department of Transportation. (4) *Traffic2* (Salinas et al., 2020): The traffic dataset contains hourly occupancy rates (between 0 and 1) of 963 car lanes of San Francisco bay area freeways. (5) *Solar-Energy* (Lai et al., 2018): Solar power production of 137 PV plants in 2006, sampled every 10 minutes. (6) *PEMS* (Liu et al., 2022): Public traffic network data from California, collected in 5-minute intervals, including subsets PEMS03, PEMS04, PEMS07, and PEMS08. (7) *PeMSD7(M)* (Chen et al., 2001): Data from the Caltrain PeMS system, containing 228 time-series collected at 5-minute intervals. (8) *PEMS-BAY* (Li et al., 2018): Traffic data from 325 sensors in the Bay Area, collected over 6 months (Jan 1, 2017, to May 31, 2017) by CalTrans Performance Measurement System (PeMS). (9) *METR-LA* (Jagadish et al., 2014): Traffic data from 207 sensors on Los Angeles County highways, collected over 4 months (Mar 1, 2012, to Jun 30, 2012).

We adopt the same data processing and train-validation-test split protocol from iTransformer (Liu et al., 2024). For long-term forecasting, the lookback length is fixed at 96 for the ECL, Traffic, and Solar-Energy datasets, with prediction lengths of {96, 192, 336, 720}. For short-term forecasting, the lookback length is also fixed at 96 for the PEMS03, PEMS04, PEMS07, PEMS08, PeMSD7(M), PEMS-BAY, METR-LA, electricity and Traffic2 datasets, with prediction lengths of {12, 24, 48, 96}. Detailed dataset statistics are provided in Table 5.

Table 5: Detailed dataset descriptions. *Dim* denotes the variable number of each dataset. *Dataset Size* denotes the total number of time points in (Train, Validation, Test) split respectively. *Prediction Length* denotes the prediction length in each dataset. *Frequency* denotes the sampling interval of time points.

| Dataset | Dim | Prediction Length | Dataset Size | Frequency | Information |
|---------|-----|-------------------|--------------|-----------|-------------|
| ECL | 321 | {96, 192, 336, 720} | (18317, 2633, 5261) | Hourly | Electricity |
| Traffic | 862 | {96, 192, 336, 720} | (12185, 1757, 3509) | Hourly | Transportation |
| Solar-Energy | 137 | {96, 192, 336, 720} | (36601, 5161, 10417) | 10min | Energy |
| electricity | 370 | {12, 24, 48, 96} | (18104, 2518, 5132) | Hourly | Electricity |
| Traffic2 | 963 | {12, 24, 48, 96} | (7284, 1045, 2101) | Hourly | Transportation |
| PEMS03 | 358 | {12, 24, 48, 96} | (15617, 5135, 5135) | 5min | Transportation |
| PEMS04 | 307 | {12, 24, 48, 96} | (10172, 3375, 3375) | 5min | Transportation |
| PEMS07 | 883 | {12, 24, 48, 96} | (16911, 5622, 5622) | 5min | Transportation |
| PEMS08 | 170 | {12, 24, 48, 96} | (10690, 3548, 3548) | 5min | Transportation |
| PeMSD7(M) | 228 | {12, 24, 48, 96} | (8763, 1256, 2523) | 5min | Transportation |
| PEMS-BAY | 325 | {12, 24, 48, 96} | (36374, 5201, 10412) | 5min | Transportation |
| METR-LA | 207 | {12, 24, 48, 96} | (23883, 3417, 6843) | 5min | Transportation |

### B.2 EXPERIMENTAL DETAILS

All the experiments are implemented in PyTorch (Paszke et al., 2019) and conducted on NVIDIA A800 GPU. Batch size is uniformly set to 32. The training epoch number is set to 15 and training process is early stopped after three epochs if there is no loss degradation on the valid set. We use L2Loss and ADAM optimizer (Kingma & Ba, 2015) with an initial learning rate of 0.001. Considering the computational efficiency and the reception field of Inception module, the number of kernels in Inception is set to 4. We employ a 5-layer DConvBlock, with the multi-dilated rates

$n$ set to $[1, 2, 3, 2, 1]$ by default and can be adjusted according to different datasets. The weight factor $\alpha$ is selected from $[0.0, 0.1, 0.3, 0.5, 0.7, 0.9, 1.0]$ to address the varying variable interactions that exist within different datasets. We repeat the experiments five times with different random seeds and report the mean results in the main text. We use the results of all baselines reported in iTransformer (Liu et al., 2024) their original papers. For models without corresponding results, we run their publicly available code using the default parameter setting. The experiment configuration with best hyperparameter is listed in Table 6 and the hyperparameter sensitivity analysis is in the Appendix B.3.

Table 6: Experiment configuration of FACT. D denotes the hidden dimension. $n$ denotes the dilation size. $\alpha$ denotes the weight factor of time domain and frequency domain. k denotes the number of kernels in Inception.

| Datasets/Configurations | D | $n$ | k | $\alpha$ | dropout | learning rate | batch size |
|---|---|---|---|---|---|---|---|
| ECL | 512 | [1,2,3,2,1] | 4 | 0.1 | 0.5 | 0.001 | 32 |
| Traffic | 1024 | [1,2,3,2,1] | 4 | 0.5 | 0.1 | 0.001 | 32 |
| Solar-Energy | 512 | [1,2,1] | 4 | 0.5 | 0.1 | 0.001 | 32 |
| electricity | 512 | [1,2,3,2,1] | 4 | 0.0 | 0.5 | 0.001 | 32 |
| Traffic2 | 1024 | [1,2,3,2,1] | 4 | 0.3 | 0.1 | 0.001 | 32 |
| PEMS03 | 512 | [1,2,3,2,1] | 4 | 0.3 | 0.1 | 0.001 | 32 |
| PEMS04 | 512 | [1,2,3,2,1] | 4 | 0.3 | 0.1 | 0.001 | 32 |
| PEMS07 | 512 | [1,2,3,4,3,2,1] | 4 | 0.3 | 0.1 | 0.001 | 32 |
| PEMS08 | 512 | [1,2,3,2,1] | 4 | 0.0 | 0.1 | 0.001 | 32 |
| PeMSD7(M) | 512 | [1,2,3,2,1] | 4 | 1.0 | 0.7 | 0.001 | 32 |
| PEMS-BAY | 512 | [1,2,3,4,3,2,1] | 4 | 0.1 | 0.7 | 0.001 | 32 |
| METR-LA | 512 | [1,2,1] | 4 | 0.9 | 0.7 | 0.001 | 32 |

## B.3 HYPERPARAMETER SENSITIVITY

We evaluate the hyperparameter sensitivity of FACT with respect to three critical factors: the weight factor $\alpha$, the number of kernels in the Inception module, the hidden dimension, and the multiple dilation rates. These experiments are conducted on four datasets with varying characteristics and scale to ensure comprehensive analysis. The hidden dimension directly influences the model's capacity and representational power. The number of kernels in the Inception and the dilation rates determine the basic reception field covered in each model layer, while the weight factor $\alpha$ controls the weight of variable interactions in time domain and the frequency domain.

**The weight factor** $\alpha$ As shown in Figure 6, we observe that different $\alpha$ yield different performance due to the different variable interactions in various datasets. In most cases, modeling both time-domain and frequency-domain variable correlations simultaneously yields superior results, as it considers both instantaneous interactions at specific data points and periodic patterns. However, in certain specific case (PEMS08 dataset), increasing $\alpha$ (i.e., increasing the weight for time domain modeling) will gradually degrade performance. We attribute this to the fact that the variable interactions in this dataset primarily involve periodic pattern co-variation, while instantaneous interactions in the time domain may introduce noise. This further underscores the importance of determining the relative weighting between time-domain and frequency-domain variable modeling.

**The number of kernels** As shown in Figure 7, increasing the number of kernels in the Inception module enlarges the reception field and tends to enhance performance. However, an overly large number of kernels will introduce optimization challenges and more training cost, which negatively impacts overall results. Considering the balance of performance and computational cost, we set the number of kernels to 4 by default.

**The hidden dimension** Similarly, as shown in Figure 8, we observe that increasing the hidden dimension generally improves forecasting performance by enabling richer feature extraction. How-

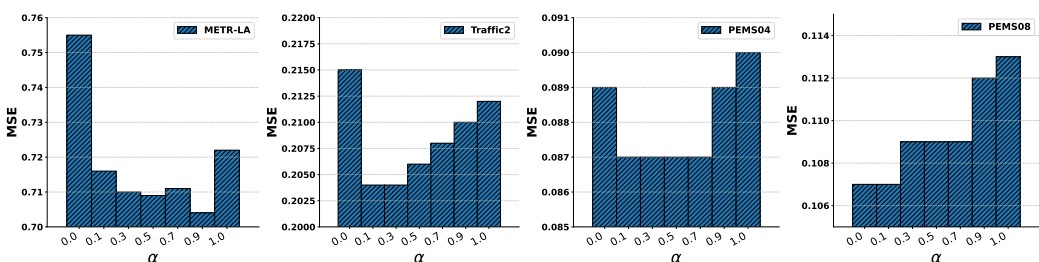

Figure 6: Hyperparameter sensitivity of $\alpha$.

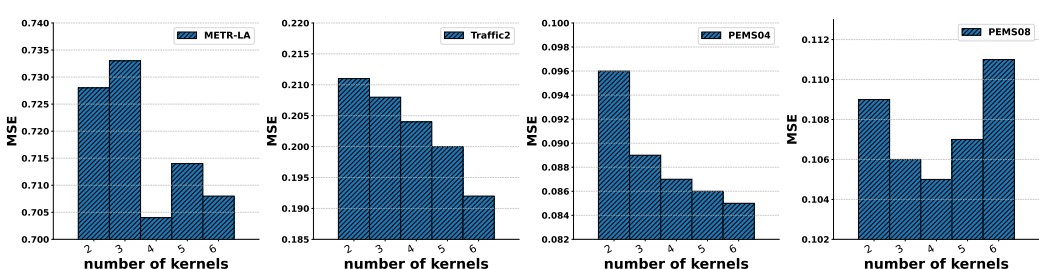

Figure 7: Hyperparameter sensitivity of the number of kernels in Inception.

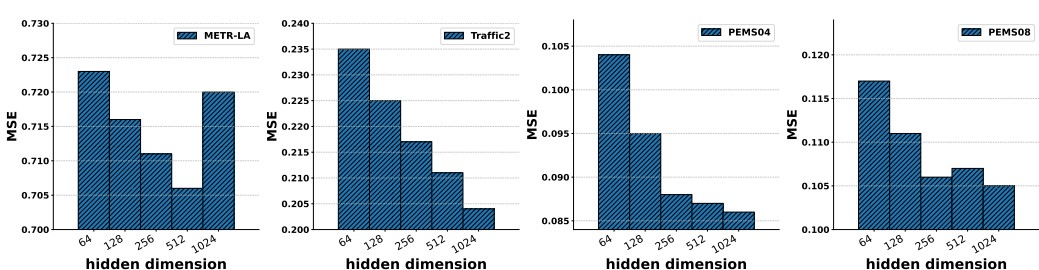

Figure 8: Hyperparameter sensitivity of the hidden dimension.

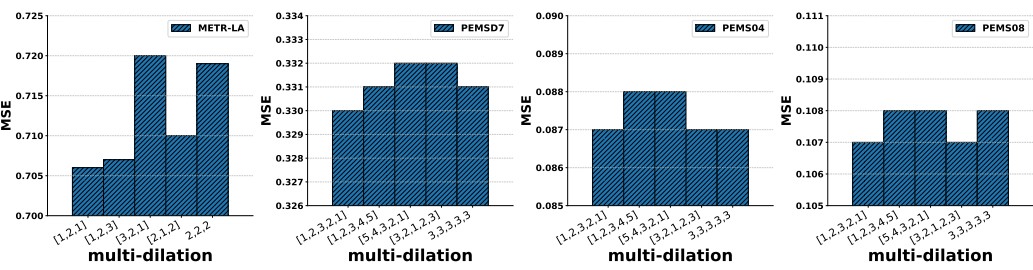

Figure 9: Hyperparameter sensitivity of the multi-dilated rates.

Table 7: Sensitivity analysis of model layers.

| Model layers | Metric | Traffic2 | PEMS04 | PEMS08 | METR-LA |
|:---:|:---:|:---:|:---:|:---:|:---:|
| 3 | MSE | 0.210 | 0.091 | 0.112 | 0.704 |
|   | MAE | 0.182 | 0.201 | 0.207 | 0.456 |
| 4 | MSE | 0.207 | 0.088 | 0.107 | 0.705 |
|   | MAE | 0.180 | 0.196 | 0.202 | 0.458 |
| 5 | MSE | 0.204 | 0.086 | 0.107 | 0.711 |
|   | MAE | 0.179 | 0.194 | 0.194 | 0.459 |
| 6 | MSE | 0.202 | 0.086 | 0.107 | 0.715 |
|   | MAE | 0.179 | 0.194 | 0.198 | 0.460 |
| 7 | MSE | 0.201 | 0.085 | 0.109 | 0.729 |
|   | MAE | 0.178 | 0.192 | 0.200 | 0.463 |

ever, excessively large hidden dimensions may cause training instability and convergence difficulties, ultimately yielding diminishing returns or even performance degradation. Considering training stability and computational cost, we set the hidden dimension to 512 by default.

**The multi-dilated rates** We adopt a palindromic multi-dilated rate configuration with increasing and then decreasing dilation factors $[1, 2, 3, 2, 1]$ as the default setting in this paper. This design ensures a balanced expansion of the reception field across layers: the dilation factors gradually increase and then decrease, preventing either small- or large-scale dependencies from dominating abruptly and reducing the number of total kernels. Consequently, the model is able to capture both local and global dependencies more smoothly, which promotes stable gradient propagation. For comparison, we also evaluate other dilation configurations, including the increasing format ($[1, 2, 3, 4, 5]$), consistent format ($[3, 3, 3, 3, 3]$), decreasing–increasing palindromic format ($[3, 2, 1, 2, 3]$), and decreasing format ($[5, 4, 3, 2, 1]$), as shown in Figure 9. Among these, the increasing–decreasing palindromic configuration consistently delivers the best overall performance in most cases.

**The number of layer** We also conduct a sensitivity analysis by varying the number of layers and the corresponding dilation rates $n$, which are directly related to the number of layers. Specifically, we evaluate several configurations: 3 layers ($n = [1, 2, 1]$), 4 layers ($n = [1, 2, 2, 1]$), 5 layers ($n = [1, 2, 3, 2, 1]$), 6 layers ($n = [1, 2, 3, 3, 2, 1]$), and 7 layers ($n = [1, 2, 3, 4, 3, 2, 1]$). As the number of layers increases from 3 to 7, the receptive field expands, enabling the model to capture broader dependency ranges. In our model, we employ a 5-layer DConvBlock with default multi-dilation rates of [1,2,3,2,1]. The results are summarized in Table 7. Increasing the number of layers enlarges the receptive field, and overall, performance tends to improve or remain stable. Different datasets exhibit varying sensitivities to receptive field size due to their distinct characteristics. On most datasets, such as Traffic2, PEMS04, and PEMS08, expanding the receptive field by increasing the number of layers leads to stable or slightly improved performance, while on METR-LA, further increasing the receptive field beyond a certain point results in performance degradation. Therefore, we set the default number of layers to 5.

These findings underscore the importance of carefully selecting an appropriate balance among these hyperparameters to maintain training efficiency and performance.

## C  ANALYSIS OF DCONVBLOCK

In this section, we analyze the key technical components of the DConvBlock in FACT, focusing on 2D modeling, multi-dilated convolution, and the choice of Inception as the convolution backbone.

**2D modeling** Specifically, the 2D modeling strategy converts the original 1D variable into 2D form and utilizes 2D convolution, effectively reducing the number of convolution layers required to achieve a global reception field and improving training efficiency. To empirically validate this, we replace the 2D modeling with a 1D convolution counterpart and report the results in Figure 10 (denoted as "+1D"). The consistent superiority of the 2D approach confirms that 1D convolutions is

difficult to cover the full reception field with an equal number of layers, thus limiting performance. Therefore, we employ 2D modeling to achieve better performance using fewer parameters and fewer model layers.

**Multi-dilated structure** In addition, we employ a multi-dilated convolution structure to further expand the reception field without increasing the number of parameters. For comparison, we evaluate the model without dilation (dilation rate = 1) and present the results in Figure 10 (denoted as "w/o dilation"). The improved performance with multi-dilated convolutions clearly indicates the critical role of a larger reception field in capturing variable interactions.

**Inception vs. standard convolution** In DConvBlock, we select the Inception as the convolution backbone. To demonstrate that our model's effectiveness derives primarily from the architecture rather than specific choice of backbone, we substitute Inception with a standard 2D convolution structure and evaluate the performance, as shown in Figure 10 (denoted as "+Conv"). Although the general convolution backbone achieves competitive results, it performs slightly worse than Inception. This suggests that the multi-dilated depth-wise design is a major factor contributing to the superior performance of FACT. More importantly, this result confirms the flexibility and extensibility of our framework, showing that it can be seamlessly integrated with various generalized convolution networks. This adaptability significantly expands the potential of convolution architectures in time series forecasting. Meanwhile, we also compare the parameter count and FLOPs of Inception and standard 2D convolution to justify the additional training cost. As shown in Table 8, the results demonstrate that FACT with the Inception module will introduce only a marginal increase in parameters and FLOPs. Since each kernel at each channel has only $1 \times H \times W$ parameters, the Inception module increases the number of kernel elements only slightly compared to a standard 2D convolution. This represents a worthwhile trade-off for improved effectiveness.

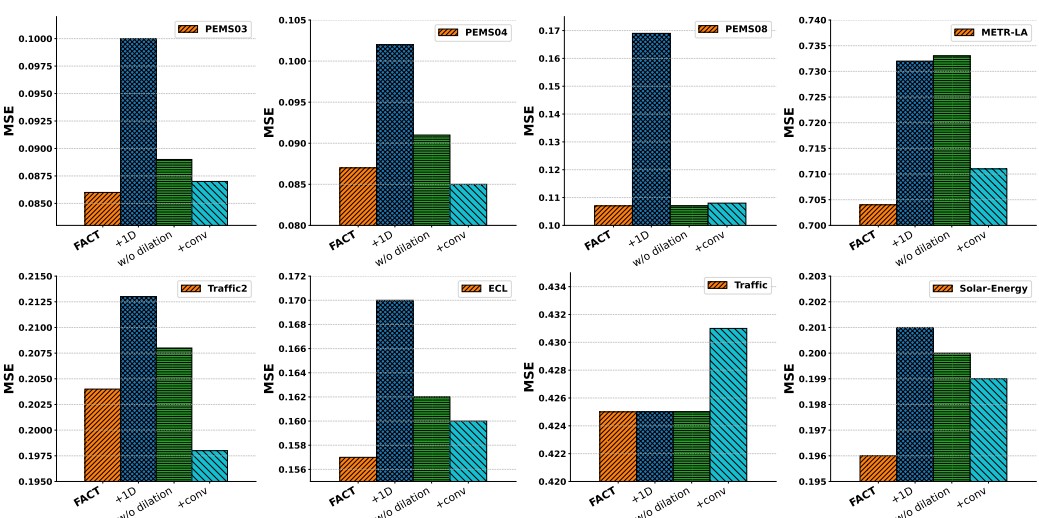

Figure 10: Analysis of DConvBlock on different datasets.

Table 8: Parameter and FLOPs comparison of FACT with Inception and standard convolution.

| Model | Metric | PEMS03 | PEMS08 | METR-LA | ECL | Traffic | Solar |
|-------|--------|--------|--------|---------|-----|---------|-------|
| FACT | parameter (M) | 21.94 | 11.07 | 3.63 | 31.05 | 21.98 | 3.63 |
|  | FLOPs (G) | 505.98 | 138.34 | 52.08 | 715.74 | 1263.74 | 33.34 |
| + Conv | parameter (M) | 21.59 | 10.89 | 3.47 | 30.43 | 21.63 | 3.47 |
|  | FLOPs (G) | 498.52 | 136.32 | 50.02 | 702.46 | 1225.12 | 32.02 |

Table 9: Analysis of different frequency modeling methods.

| Datasets | | PEMS03 | | PEMS04 | | PEMS08 | | METR-LA | | Traffic2 | | | ECL | | Traffic | | Solar-Energy | |
|---|---|---|---|---|---|---|---|---|---|---|---|---|---|---|---|---|---|---|
| Metric | | MSE | MAE | MSE | MAE | MSE | MAE | MSE | MAE | MSE | MAE | | MSE | MAE | MSE | MAE | MSE | MAE |
| **FACT** | 12 | 0.058 | 0.159 | 0.067 | 0.169 | 0.068 | 0.167 | 0.425 | 0.324 | 0.181 | 0.165 | 96 | 0.131 | 0.227 | 0.395 | 0.261 | 0.175 | 0.220 |
| | 24 | 0.072 | 0.178 | 0.078 | 0.185 | 0.086 | 0.185 | 0.603 | 0.404 | 0.196 | 0.173 | 192 | 0.149 | 0.244 | 0.413 | 0.270 | 0.197 | 0.259 |
| | 48 | 0.094 | 0.205 | 0.093 | 0.202 | 0.117 | 0.211 | 0.791 | 0.503 | 0.213 | 0.185 | 339 | 0.163 | 0.260 | 0.430 | 0.277 | 0.198 | 0.265 |
| | 96 | 0.123 | 0.236 | 0.110 | 0.222 | 0.160 | 0.238 | 0.998 | 0.596 | 0.228 | 0.195 | 720 | 0.188 | 0.286 | 0.464 | 0.296 | 0.214 | 0.272 |
| | avg | **0.086** | 0.194 | **0.087** | **0.194** | **0.107** | **0.200** | **0.704** | **0.456** | **0.204** | 0.179 | avg | **0.157** | **0.254** | 0.425 | 0.276 | **0.196** | **0.254** |
| **+ concat** | 12 | 0.059 | 0.162 | 0.068 | 0.170 | 0.167 | 0.181 | 0.425 | 0.315 | 0.181 | 0.164 | 96 | 0.135 | 0.232 | 0.393 | 0.262 | 0.180 | 0.238 |
| | 24 | 0.072 | 0.178 | 0.078 | 0.184 | 0.202 | 0.192 | 0.622 | 0.401 | 0.196 | 0.172 | 192 | 0.157 | 0.252 | 0.414 | 0.271 | 0.201 | 0.258 |
| | 48 | 0.096 | 0.207 | 0.095 | 0.207 | 0.201 | 0.218 | 0.808 | 0.517 | 0.212 | 0.184 | 339 | 0.165 | 0.261 | 0.431 | 0.278 | 0.210 | 0.269 |
| | 96 | 0.124 | 0.239 | 0.112 | 0.225 | 0.215 | 0.230 | 0.995 | 0.605 | 0.229 | 0.196 | 720 | 0.187 | 0.284 | 0.466 | 0.297 | 0.212 | 0.269 |
| | avg | 0.087 | 0.196 | 0.088 | 0.196 | 0.196 | 0.205 | 0.712 | 0.459 | 0.204 | 0.179 | avg | 0.161 | 0.257 | 0.426 | 0.277 | 0.200 | 0.258 |
| **+ share** | 12 | 0.181 | 0.348 | 0.210 | 0.376 | 0.584 | 0.620 | 0.427 | 0.332 | 0.310 | 0.330 | 96 | 0.226 | 0.341 | 0.454 | 0.328 | 0.270 | 0.409 |
| | 24 | 0.188 | 0.348 | 0.138 | 0.263 | 1.003 | 0.809 | 0.600 | 0.404 | 0.298 | 0.308 | 192 | 0.213 | 0.331 | 0.497 | 0.363 | 0.270 | 0.387 |
| | 48 | 0.210 | 0.363 | 0.166 | 0.320 | 0.956 | 0.807 | 0.803 | 0.509 | 0.299 | 0.301 | 339 | 0.267 | 0.380 | 0.531 | 0.380 | 0.303 | 0.430 |
| | 96 | 0.180 | 0.325 | 0.155 | 0.296 | 1.157 | 0.881 | 1.002 | 0.600 | 0.331 | 0.328 | 720 | 0.343 | 0.439 | 0.580 | 0.400 | 0.328 | 0.447 |
| | avg | 0.189 | 0.346 | 0.167 | 0.313 | 0.927 | 0.779 | 0.708 | 0.461 | 0.310 | 0.316 | avg | 0.262 | 0.372 | 0.515 | 0.367 | 0.292 | 0.418 |
| **+ complex** | 12 | 0.058 | 0.159 | 0.067 | 0.170 | 0.070 | 0.171 | 0.422 | 0.310 | 0.185 | 0.161 | 96 | 0.135 | 0.230 | 0.389 | 0.257 | 0.182 | 0.241 |
| | 24 | 0.071 | 0.176 | 0.078 | 0.183 | 0.089 | 0.191 | 0.612 | 0.399 | 0.199 | 0.169 | 192 | 0.153 | 0.246 | 0.411 | 0.267 | 0.199 | 0.260 |
| | 48 | 0.095 | 0.205 | 0.094 | 0.205 | 0.118 | 0.218 | 0.800 | 0.533 | 0.214 | 0.182 | 339 | 0.168 | 0.263 | 0.431 | 0.275 | 0.204 | 0.262 |
| | 96 | 0.122 | 0.236 | 0.112 | 0.224 | 0.160 | 0.248 | 1.013 | 0.600 | 0.229 | 0.192 | 720 | 0.207 | 0.302 | 0.465 | 0.294 | 0.221 | 0.280 |
| | avg | **0.086** | 0.194 | **0.087** | 0.195 | 0.109 | 0.207 | 0.711 | 0.460 | 0.206 | 0.176 | avg | 0.165 | 0.260 | **0.423** | **0.273** | 0.201 | 0.260 |
| **+ real-imag** | 12 | 0.058 | 0.195 | 0.067 | 0.169 | 0.072 | 0.172 | 0.424 | 0.323 | 0.184 | 0.161 | 96 | 0.134 | 0.228 | 0.389 | 0.257 | 0.174 | 0.234 |
| | 24 | 0.071 | 0.176 | 0.077 | 0.183 | 0.092 | 0.193 | 0.598 | 0.399 | 0.198 | 0.168 | 192 | 0.151 | 0.244 | 0.412 | 0.267 | 0.207 | 0.260 |
| | 48 | 0.093 | 0.203 | 0.093 | 0.204 | 0.121 | 0.221 | 0.803 | 0.507 | 0.215 | 0.181 | 339 | 0.168 | 0.263 | 0.430 | 0.275 | 0.207 | 0.265 |
| | 96 | 0.122 | 0.236 | 0.112 | 0.225 | 0.165 | 0.250 | 1.001 | 0.595 | 0.229 | 0.191 | 720 | 0.189 | 0.286 | 0.470 | 0.294 | 0.204 | 0.263 |
| | avg | **0.086** | **0.193** | **0.087** | 0.195 | 0.112 | 0.209 | 0.706 | **0.456** | 0.206 | **0.175** | avg | 0.160 | 0.255 | 0.425 | **0.273** | 0.198 | 0.255 |

## D ANALYSIS OF FREQUENCY MODELING

**Frequency modeling methods** In this section, we analyze our frequency-domain modeling approach, which captures variable interactions by separately modeling the amplitude and phase of frequency components. We compare it with other four representative alternatives: *parameter sharing* employs the same model parameters for both the amplitude and phase; *concatenation* concatenates the amplitude and phase parts before applying a unified model; *complex convolution* operates directly in the complex domain, where weights, inputs, and outputs are complex valued; *real-imag* models the real and imaginary parts separately rather than amplitude and phase. Table 9 shows the results of performance comparison on diverse datasets. Although *complex convolution* and *real-imag* perform competitively, our separate modeling of amplitude and phase can achieve the better performance across most datasets. To further assess the practical benefits of the amplitude and phase modeling over the Real/Imag alternative, we conducted a convergence analysis under identical parameter settings. While the Amplitude/Phase modeling introduces additional non-linear operations (arctan, cos, sin), it does not increase the number of trainable parameters. We evaluated both methods on four datasets and visualized their test performance throughout training. As shown in Figure 11, two limitations of the Real/Imag alternative are evident: (1) it often fails to converge to the global optimum, negatively impacting final performance; and (2) even when near-optimal solutions are reached, Real/Imag exhibits greater instability and larger performance fluctuations compared to Amplitude/Phase. Therefore, we adopt the Amplitude/Phase modeling in practice.

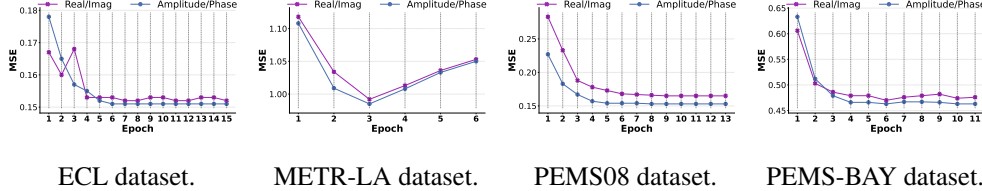

| ECL dataset. | METR-LA dataset. | PEMS08 dataset. | PEMS-BAY dataset. |
|---|---|---|---|

Figure 11: Test performance comparison with real-imag method in the training process.

**Compared with other frequency modeling models** To further evaluate the effectiveness of our proposed frequency modeling design, we conduct a comprehensive comparison with several representative frequency-based models, including FreDF (Wang et al., 2025a), FreTS (Yi et al., 2023), FilterNet (Yi et al., 2024), FITS (Xu et al., 2024), and Fredformer (Piao et al., 2024), all of which are well-known for capturing frequency-domain patterns in time series data. FreDF introduced fre-

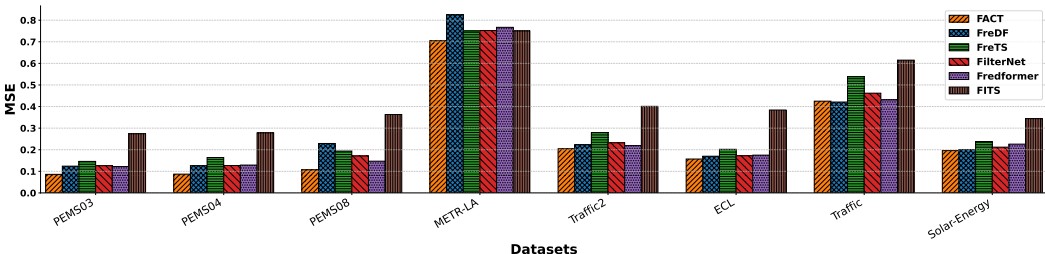

Figure 12: Compared with other frequency-based models.

quency forecast error loss, FreTS employed frequency MLP to learn temporal dependencies and channel dependencies, FilterNet designed frequency filters to extract key informative temporal patterns by selectively passing or attenuating certain components of time series signals, FITS argued that time series can be manipulated through interpolation in the complex frequency domain with few parameters, and Fredformer proposed to use channel-wise attention in frequency domain to model the channel dependencies. In contrast, our FACT focuses on capturing fine-grained variable interactions by modeling the distinct dependencies of variables at each frequency components. As shown in Figure 12, FACT consistently outperforms these models across all datasets. This consistent superiority not only demonstrates the effectiveness of our frequency modeling approach but also underscores the advantage of explicitly capturing fine-grained cross-variable interactions from the frequency-domain perspective.

# E  MODEL ANALYSIS

## E.1  EMBEDDING ANALYSIS

Given a multivariate time series $\mathbf{X} \in \mathbb{R}^{C \times T}$ consisting of $C$ variables, we first project each length-$T$ variable into a high-dimensional embedding space of dimension $D$ to get its length-$D$ embedding, resulting in $\mathbf{X}_e \in \mathbb{R}^{C \times D}$. We then apply the Discrete Fourier Transform (DFT) to $\mathbf{X}_e$ along the embedding dimension $D$ to capture the variable interactions in frequency domain.

It is important to note that the embedding is computed on a per-variable basis; specifically, each $T$-length variable is projected into a $D$-length embedding vector, ensuring that temporal information is implicitly preserved within the embedding space. Compared to applying DFT directly on $\mathbf{X}$ along the temporal dimension $T$, our approach introduces a learnable projection step. This provides the flexibility to adapt to complex temporal patterns within a high-dimensional latent space, yielding more expressive representations for subsequent frequency modeling. Meanwhile, we also compared FACT against a variant that applies DFT directly along the temporal dimension $T$ of the original series $\mathbf{X}$ without embedding. The results, summarized in Table 10, show that FACT consistently outperforms the direct DFT variant without embedding across all datasets, validating the effectiveness of our design choice.

Table 10: Comparison of FACT with embedding and the variant applying DFT directly on the original series without embedding.

| Model | Metric | PEMS03 | PEMS04 | PEMS08 | METR-LA | Traffic2 | ECL | Traffic | Solar |
|---|---|---|---|---|---|---|---|---|---|
| FACT | MSE | **0.086** | **0.087** | **0.107** | **0.704** | **0.204** | **0.157** | **0.425** | **0.196** |
| | MAE | **0.194** | **0.194** | **0.200** | **0.456** | **0.179** | **0.254** | **0.276** | **0.254** |
| w/o Embedding | MSE | 0.101 | 0.102 | 0.143 | 0.724 | 0.216 | 0.177 | 0.467 | 0.203 |
| | MAE | 0.213 | 0.216 | 0.236 | 0.475 | 0.200 | 0.274 | 0.312 | 0.269 |

## E.2  2D VARIABLE RECONSTRUCTION STRATEGIES

For a multivariate time series with $C$ variables, we reshape the input into a 2D grid of size $H \times W$ using row-major order, preserving the original sequence of variables as provided in the input.

Concretely, the first $W$ variables are placed in the first row, the next $W$ in the second row, and so on. If $C$ is not divisible by $W$, we pad the last row by repeating the first few variables from the beginning of the sequence (former-padding) to complete the $H \times W$ grid.

**Physical topology** We do not assume any prior knowledge about the order among the $C$ variables, such as physical adjacency, when reshaping them into an $H \times W$ grid for 2D variable reconstruction. Our method is designed for time series data, regardless of whether the variables have inherent physical topology or not. This design choice allows our method to be broadly applicable to various multivariate time series forecasting tasks without being constrained by the need for spatial information.

Meanwhile, we conduct experiments to evaluate the impact of incorporating physical topology by comparing with (1) Clustering-based layout, where variables are grouped by correlation strength in 1D and then reshaped into a 2D grid, positioning highly correlated variables closer together; (2) MDS-based layout, where Multidimensional Scaling (MDS) projects variables into 2D space, preserving correlation-based proximity. While clustering-based layouts partially capture topology, MDS-based layouts offer a more rigorous representation. The results in Table 11 show that the performance of these two variant layouts is comparable to our original FACT model, validating that our model does not rely on physical topology for effective forecasting, and thus is generalizable to various datasets without inherent variable order.

Table 11: Comparison of incorporating physical topology.

| Model | Metric | PEMS03 | PEMS04 | PEMS08 | METR-LA | Traffic2 | PEMSD7 | PEMS-BAY |
|---|---|---|---|---|---|---|---|---|
| FACT | MSE | 0.086 | 0.087 | 0.107 | 0.704 | 0.204 | 0.330 | 0.372 |
| | MAE | 0.194 | 0.194 | 0.200 | 0.456 | 0.179 | 0.316 | 0.267 |
| + Cluster | MSE | 0.087 | 0.085 | 0.113 | 0.717 | 0.204 | 0.328 | 0.370 |
| | MAE | 0.195 | 0.191 | 0.204 | 0.458 | 0.179 | 0.315 | 0.271 |
| + MDS | MSE | 0.086 | 0.085 | 0.106 | 0.699 | 0.203 | 0.332 | 0.371 |
| | MAE | 0.194 | 0.192 | 0.198 | 0.457 | 0.179 | 0.315 | 0.271 |

**Variable permutation** Moreover, to assess the effect of variable permutation, we compare model performance when the variables are randomly shuffled (e.g., swapping variable 5 and variable 50). Additionally, to further demonstrate the robustness of our approach, we evaluate several alternative 1D-to-2D variable reshaping methods, including column-major, Z-order, snake, spiral, and Hilbert curve layouts. The results (summarized in Table 12) show that random permutation of variable order have minimal impact on model performance, indicating that our method remains stable and effective even when variable indices are swapped. Similarly, performance differences across various 2D reshaping methods are negligible, further validating that our approach does not rely on specific variable adjacency or arrangement for accurate forecasting.

Table 12: Comparison of different 1D-to-2D variable reshaping methods.

| Model | Metric | PEMS03 | PEMS04 | PEMS08 | METR-LA | Traffic2 | PEMSD7 | PEMS-BAY |
|---|---|---|---|---|---|---|---|---|
| FACT (row-major) | MSE | 0.086 | 0.087 | 0.107 | 0.704 | 0.204 | 0.330 | 0.372 |
| | MAE | 0.194 | 0.194 | 0.200 | 0.456 | 0.179 | 0.316 | 0.267 |
| + random | MSE | 0.087 | 0.088 | 0.110 | 0.706 | 0.206 | 0.331 | 0.369 |
| | MAE | 0.196 | 0.196 | 0.202 | 0.458 | 0.180 | 0.336 | 0.270 |
| + column-major | MSE | 0.086 | 0.086 | 0.106 | 0.709 | 0.204 | 0.329 | 0.371 |
| | MAE | 0.194 | 0.194 | 0.201 | 0.457 | 0.179 | 0.315 | 0.268 |
| + z-order | MSE | 0.087 | 0.086 | 0.107 | 0.706 | 0.206 | 0.330 | 0.377 |
| | MAE | 0.195 | 0.194 | 0.202 | 0.456 | 0.180 | 0.316 | 0.270 |
| + snake | MSE | 0.087 | 0.087 | 0.104 | 0.707 | 0.205 | 0.330 | 0.369 |
| | MAE | 0.194 | 0.195 | 0.199 | 0.456 | 0.179 | 0.316 | 0.267 |
| + spiral | MSE | 0.086 | 0.087 | 0.108 | 0.705 | 0.206 | 0.332 | 0.375 |
| | MAE | 0.195 | 0.195 | 0.201 | 0.454 | 0.179 | 0.317 | 0.270 |
| + hilbert | MSE | 0.086 | 0.087 | 0.105 | 0.715 | 0.205 | 0.332 | 0.380 |
| | MAE | 0.194 | 0.195 | 0.197 | 0.454 | 0.179 | 0.318 | 0.271 |

**Padding strategies** Additionally, we evaluated the impact of different padding strategies when $C$ is not divisible by $W$, with results summarized in Table 13. Specifically, we compared Latter-padding, which pads the grid by repeating the last few variables, and Random-padding, which pads by randomly selecting variables from the original $C$ variables. The results demonstrate that these padding methods yield comparable performance, indicating that the choice of padding strategy does not significantly affect model effectiveness in this setting.

Table 13: Comparison of different padding methods.

| Model | Metric | PEMS03 | PEMS04 | PEMS08 | METR-LA | Traffic2 | ECL | Traffic | Solar |
|---|---|---|---|---|---|---|---|---|---|
| FACT (former-padding) | MSE | 0.086 | 0.087 | 0.107 | 0.704 | 0.204 | 0.157 | 0.425 | 0.196 |
| | MAE | 0.194 | 0.194 | 0.200 | 0.456 | 0.179 | 0.254 | 0.276 | 0.254 |
| + Latter-padding | MSE | 0.086 | 0.086 | 0.106 | 0.709 | 0.205 | 0.158 | 0.428 | 0.197 |
| | MAE | 0.194 | 0.194 | 0.202 | 0.457 | 0.179 | 0.255 | 0.276 | 0.256 |
| + Random-padding | MSE | 0.087 | 0.087 | 0.105 | 0.707 | 0.203 | 0.158 | 0.427 | 0.198 |
| | MAE | 0.195 | 0.195 | 0.199 | 0.457 | 0.180 | 0.256 | 0.277 | 0.257 |

### E.3 ENHANCING OTHER FORECASTING BASELINES

To demonstrate the effectiveness of the multi-dilated depth-wise Inception module (MDInception) in capturing variable interactions, we integrate this module into several representative baselines: the variable-dependent TimesNet, the variable-independent PatchTST, and the attention-based iTransformer. The results, presented in Figure 13, clearly show that MDInception significantly enhances the performance of all evaluated baselines. Notably, the improvement is most pronounced for PatchTST, which treats multivariate time series as multiple independent univariate series and does not explicitly model variable relationships. Incorporating MDInception into PatchTST leads to a substantial reduction in Mean Squared Error (MSE) by 15% to 30% across most datasets. These findings validate that the multi-dilated depth-wise Inception module effectively captures dynamic and fine-grained interactions among variables at each granularity. By leveraging fine-grained information from other variables, MDInception enriches variable representations, thereby substantially boosting the overall predictive capability of diverse multivariate time series models.

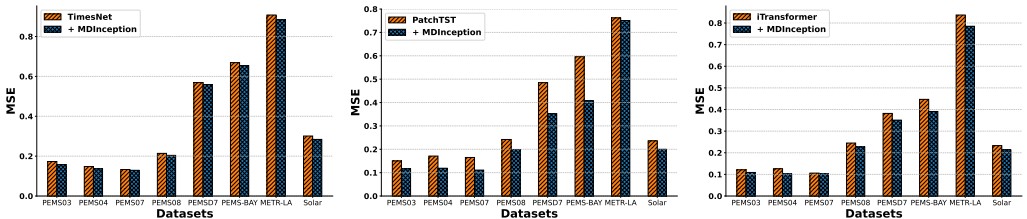

Figure 13: Performance improvements of plugging in the multi-dilated depth-wise Inception module into other models.

### E.4 COMPARISON WITH GNN BASELINES.

We also conduct experiments to compare with several representative GNN models: MTGNN (Wu et al., 2020), DCRNN (Li et al., 2018), AGCRN (Bai et al., 2020), Graph WaveNet (Wu et al., 2019), STGCN (Yu et al., 2017), and the latest TimeFilter (Hu et al., 2025). As shown in Table 14, we can note that our method achieves superior performance compared to all these GNN baselines, including the latest strong GNN model TimeFilter. These results indicate that our approach is effective even when compared against latest GNN models, and it validates the effectiveness of our proposed techniques in the paper.

### E.5 COMPARED WITH TIMEFILTER AND MODERNTCN

We also compare the training cost of our FACT with TimeFilter (Hu et al., 2025) and ModernTCN (Luo & Wang, 2024). TimeFilter employs graph-based structures to capture interactions

Table 14: Compared with other GNN baselines.

| Model | Metric | PEMS03 | PEMS04 | PEMS07 | PEMS08 | METR-LA | PEMSD7 | PEMS-BAY |
|---|---|---|---|---|---|---|---|---|
| **FACT** | MSE | **0.086** | **0.087** | **0.073** | **0.107** | **0.704** | **0.330** | **0.372** |
| | MAE | **0.194** | **0.194** | **0.170** | **0.200** | **0.456** | **0.316** | **0.267** |
| DCRNN | MSE | 0.273 | 0.243 | 0.479 | 0.542 | 0.915 | 0.644 | 0.704 |
| | MAE | 0.380 | 0.360 | 0.358 | 0.426 | 0.602 | 0.529 | 0.479 |
| Graph WaveNet | MSE | 0.186 | 0.184 | 0.159 | 0.261 | 0.884 | 0.443 | 0.499 |
| | MAE | 0.289 | 0.290 | 0.266 | 0.304 | 0.551 | 0.392 | 0.345 |
| MTGNN | MSE | 0.198 | 0.195 | 0.174 | 0.294 | 0.895 | 0.532 | 0.609 |
| | MAE | 0.298 | 0.302 | 0.281 | 0.327 | 0.577 | 0.439 | 0.393 |
| AGCRN | MSE | 0.157 | 0.150 | 0.177 | 0.266 | 0.748 | 0.442 | 0.510 |
| | MAE | 0.269 | 0.267 | 0.275 | 0.292 | 0.501 | 0.372 | 0.360 |
| STGCN | MSE | 0.101 | 0.092 | 0.094 | 0.208 | 0.732 | 0.343 | 0.396 |
| | MAE | 0.208 | 0.203 | 0.182 | 0.240 | 0.473 | 0.321 | 0.291 |
| TimeFilter | MSE | 0.108 | 0.114 | 0.101 | 0.139 | 0.819 | 0.498 | 0.547 |
| | MAE | 0.217 | 0.226 | 0.207 | 0.230 | 0.493 | 0.404 | 0.358 |

among variables, thereby enabling fine-grained modeling. ModernTCN also adopts convolution neural network backbone to extract inter-variable correlations. In Table 1 and Table 2, we demonstrated that the superior performance of our FACT compared with TimeFilter and ModernTCN. Here, we further compare the training costs of them to provide a comprehensive evaluation.

As illustrated in Figure 14, our proposed FACT consistently demonstrates superior computational efficiency across all evaluated datasets. This advantage holds for datasets with relatively fewer variables, such as Solar-Energy and METR-LA, as well as for large-scale datasets with substantially more variables, including PEMS07, Traffic, and Traffic2. The consistent performance and efficiency across different scales of variable dimensionality highlights the robustness and scalability of FACT in handling both small- and large-scale multivariate forecasting tasks.

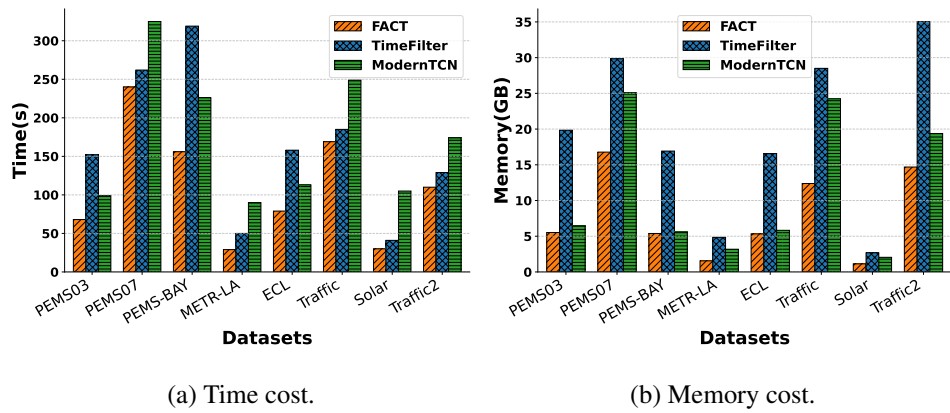

(a) Time cost.                    (b) Memory cost.

Figure 14: Efficiency comparison with TimeFilter and ModernTCN.

### E.6 DEPTH-WISE CONVOLUTION WITH EXPLICIT MODELING BETWEEN GRANULARITIES

In depth-wise architecture, there are often alternative approaches to modeling the relationships among different channels, such as Squeeze-and-Excite (SENet) (Hu et al., 2018) block. Therefore, we also evaluated our method FACT with its variant incorporating the SENet block. The results, summarized in Table 15, show that FACT with and without the SENet block achieve comparable performance across all datasets. We attribute this to our point-wise FFN module following the depth-wise Inception block, which already effectively models relationships between channels. Consequently, adding the SENet block does not yield additional performance gains.

Table 15: Compared with combining SENet block.

| Model | Metric | PEMS03 | PEMS04 | PEMS08 | METR-LA | Traffic2 | ECL | Traffic | Solar | PEMSD7 | PEMS-BAY |
|-------|--------|--------|--------|--------|---------|----------|------|---------|-------|--------|----------|
| FACT | MSE | 0.086 | 0.087 | 0.107 | 0.704 | 0.204 | 0.157 | 0.425 | 0.196 | 0.330 | 0.372 |
|      | MAE | 0.194 | 0.194 | 0.200 | 0.456 | 0.179 | 0.254 | 0.276 | 0.254 | 0.316 | 0.267 |
| + SENet | MSE | 0.088 | 0.086 | 0.106 | 0.718 | 0.201 | 0.158 | 0.430 | 0.199 | 0.330 | 0.371 |
|        | MAE | 0.196 | 0.194 | 0.202 | 0.473 | 0.178 | 0.255 | 0.277 | 0.261 | 0.316 | 0.270 |

### E.7   LONGER INPUT LENGTH

Longer input lengths provide models with more historical information, which, when effectively leveraged, can significantly improve forecasting accuracy. To comprehensively evaluate the ability of FACT to utilize extended historical context, we conduct experiments varying the input sequence lengths over the set {96, 192, 336, 576, 720}. For a fair comparison, we select two benchmark datasets: ECL and Traffic. We also include six recent state-of-the-art baselines that employ different strategies for modeling variable relationships, including the CNN-based ModernTCN, attention-based iTransformer, graph-based TimeFilter, Mamba-based TimePro, variable-dependent TimesNet, and variable-independent PatchTST.

Figures 15 depict the forecasting performance of all models under varying lookback lengths on ECL and Traffic datasets. Our results demonstrate that FACT consistently achieves state-of-the-art accuracy across all input lengths on both datasets. This stability and robustness indicate that FACT effectively leverages extended historical information to improve prediction quality, outperforming other models regardless of the amount of past data available. These findings validate the strong capability of FACT in capturing complex, long-range dependencies inherent in multivariate time series, making it highly suitable for real-world forecasting tasks where the length of historical data can vary significantly.

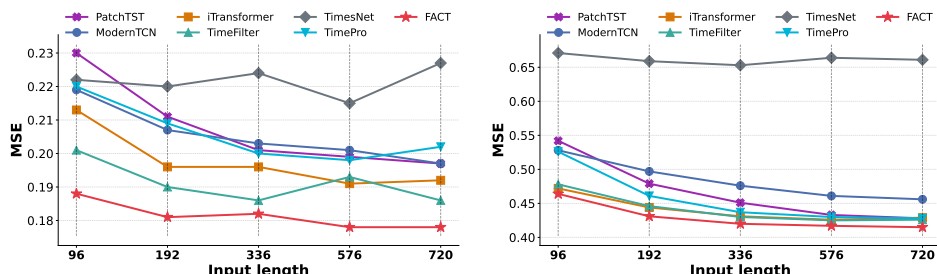

Figure 15: Performance comparison with longer input length on ECL dataset (left) and Traffic dataset (right).

## F   THE USE OF LARGE LANGUAGE MODELS (LLMS)

Large Language Models (LLMs) are only used to correct grammatical errors and to polish the syntax of the article.

## G   VISUALIZATION

### G.1   ACTIVATION MAPS VISUALIZATION

To further enhance interpretability and verify whether the model truly captures meaningful variable interactions, we provide the visualization analysis.

Specifically, we visualize the overall correlation matrix of all variables in the time domain alongside the corresponding activation map of our time-domain convolution, as well as the variable amplitude-correlation matrix and variable phase-correlation matrix in the frequency domain together with the activation maps of amplitude and phase interactions from our frequency-domain convolution. We

include three representative datasets and report the results in Figure 16(a), Figure 17(a), Figure 18(a). These visualizations show that variable pairs with strong correlations exhibit strong responses in the learned activation maps, while weakly correlated pairs show weak activation responses. This demonstrates that our model effectively captures cross-variable interactions, even though the 2D grid is not tied to a true physical topology. These results provide direct evidence for the interpretability and reliability of our variable interaction modeling, and further show that our model can perform multivariate time series forecasting on real-world datasets without requiring additional information about physical topology among variables.

Further, we also visualize variable correlation and activation maps across multiple granularities, including different channels in the time domain and each frequencies in the frequency domain. At each granularity, we present the variable correlation matrix in the time domain alongside the corresponding activation map, and the amplitude-correlation and phase-correlation matrices in the frequency domain together with activation maps capturing amplitude and phase interactions. For clearer illustration, we include three different datasets, two different granularities and ten random variables for visualization. The results are reported in Figure 16(b), Figure 17(b), Figure 18(b). Our findings show that, for the same dataset, variable correlations can differ substantially across granularities, which aligns with our model design that explicitly captures dynamic fine-grained variable interactions. Moreover, within each granularity, strongly correlated variables consistently exhibit higher activation responses, while weakly correlated variables show weaker responses, both in the time and frequency domains. This consistent pattern demonstrates that our convolution structure effectively captures fine-grained cross-variable interactions.

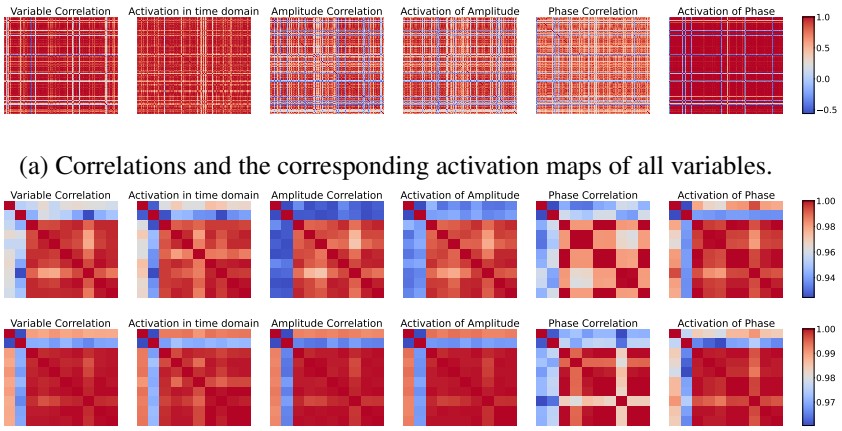

(a) Correlations and the corresponding activation maps of all variables.

(b) Correlations and activation maps of two granularities (the first line and second line).

Figure 16: The visualizaiton of variable correlation and activation maps on PEMS-BAY dataset.

## G.2 SHOWCASES

To further evaluate the forecasting performance and illustrate the effectiveness of our model, we present representative prediction showcases from the test dataset. We compare the predictions of FACT against several strong baselines, including CycleNet, TimePro, TimeFilter, iTransformer, ModernTCN, TimesNet and PatchTST, on several datasets. These baselines adopt a variety of strategies to capture inter-variable relationships, which include attention-based, graph-based, convolution-based, Mamba-based, variable dependent-based, and variable independent-based methods, thus providing a comprehensive and diverse comparison.

As illustrated in Figures 19, Figures 20, Figures 21, Figures 22, Figures 23 and Figures 24, FACT demonstrates superior ability to fit the overall trends of the time series and effectively captures temporal patterns and changes across all datasets. This highlights the model's capability to deliver more accurate and stable predictions compared to existing approaches.

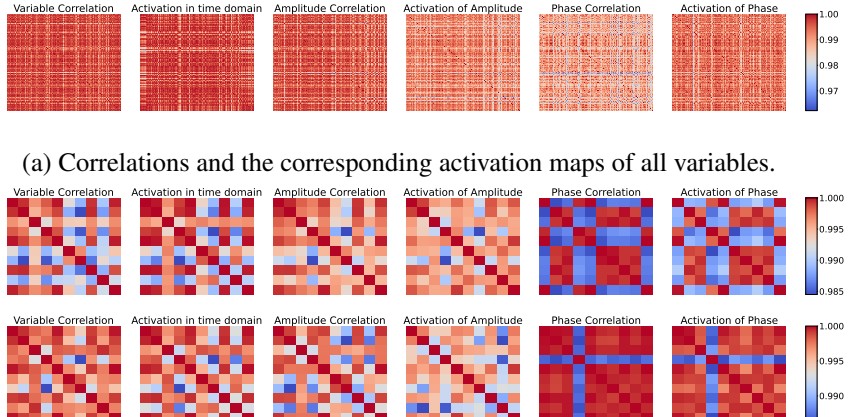

(a) Correlations and the corresponding activation maps of all variables.

(b) Correlations and activation maps of two granularities (the first line and second line).

Figure 17: The visualizaiton of variable correlation and activation maps on PEMS03 dataset.

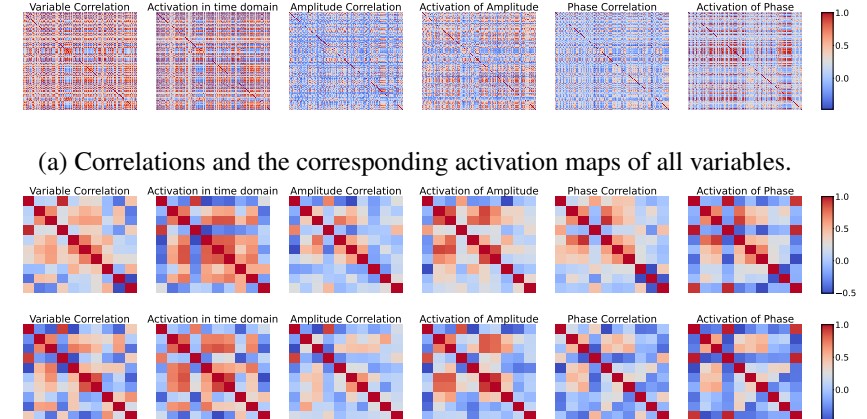

(a) Correlations and the corresponding activation maps of all variables.

(b) Correlations and activation maps of two granularities (the first line and second line).

Figure 18: The visualizaiton of variable correlation and activation maps on METR-LA dataset.

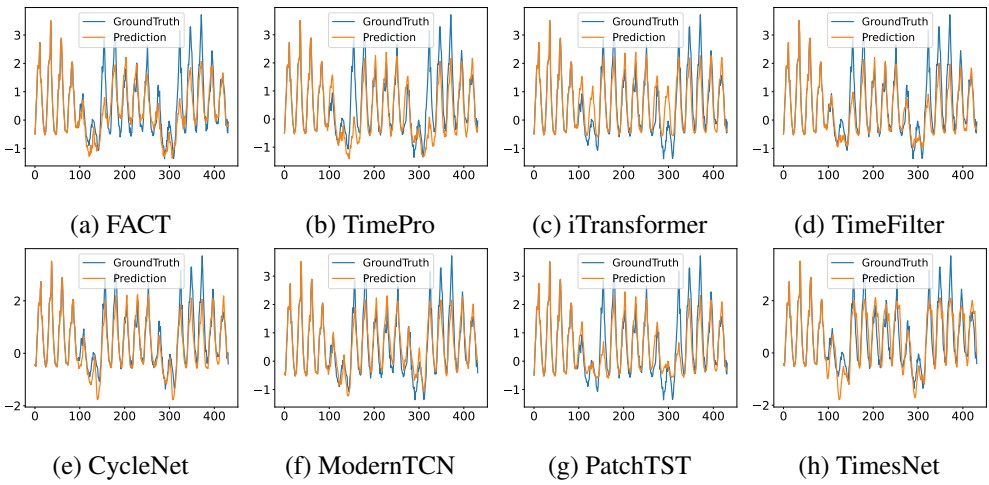

(a) FACT      (b) TimePro      (c) iTransformer      (d) TimeFilter

(e) CycleNet      (f) ModernTCN      (g) PatchTST      (h) TimesNet

Figure 19: Showcases on ECL dataset. The lookback length is 96 and prediction length is 336.

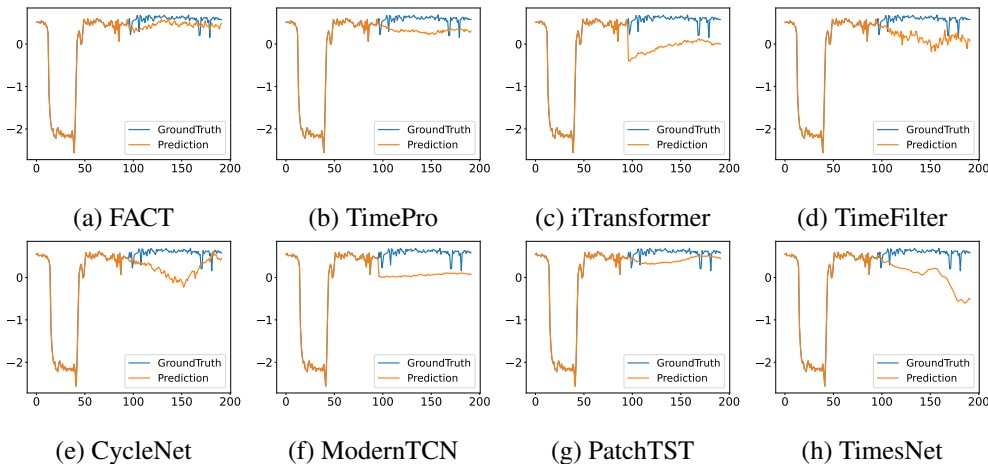

Figure 20: Showcases on METR-LA dataset. The lookback length is 96 and prediction length is 96.

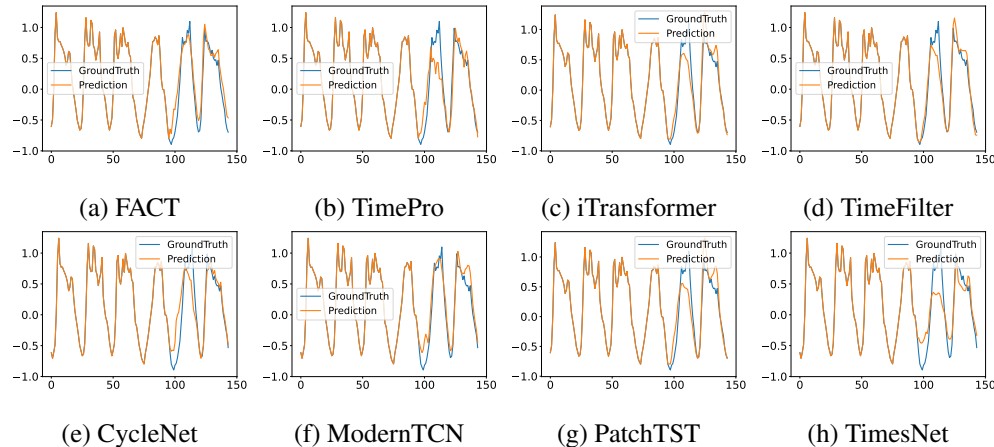

Figure 21: Showcases on Traffic2 dataset. The lookback length is 96 and prediction length is 48.

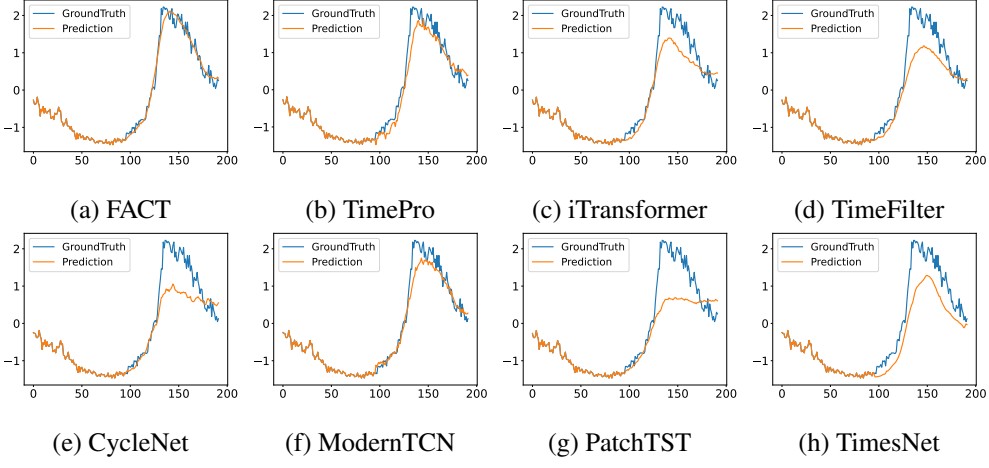

Figure 22: Showcases on PEMS03 dataset. The lookback length is 96 and prediction length is 96.

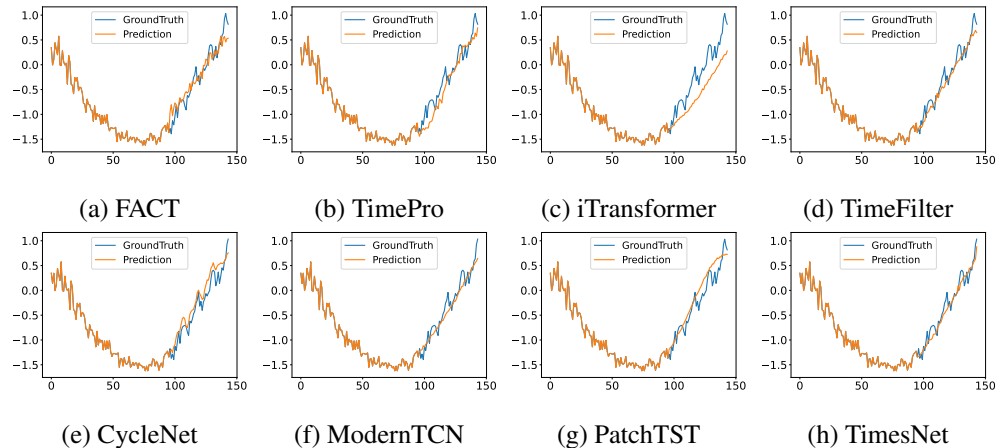

Figure 23: Showcases on PEMS04 dataset. The lookback length is 96 and prediction length is 48.

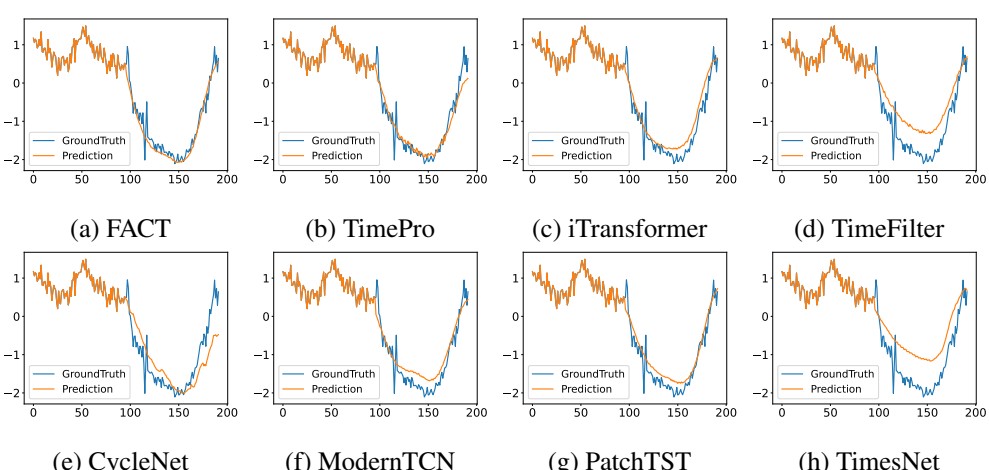

Figure 24: Showcases on PEMS07 dataset. The lookback length is 96 and prediction length is 96.

