# OpenReview forum: "FACT: Fine-grained Across-variable Convolution for Multivariate Time Series Forecasting"
_ICLR.cc/2026/Conference — ICLR 2026 Poster_

### Official Review · Reviewer_5JYU · 2025-10-15

**Soundness:** 3
**Presentation:** 2
**Contribution:** 2
**Rating:** 4
**Confidence:** 4

**Summary:**

This paper proposes FACT, which addresses the limitation where the inter-variable dependency should be modelled dynamically over time. The core design of the paper enables: (1) dependency leveraged across both the time and frequency domains; (2) dependency leveraged at different reception fields; (3) Similar to TimesNet, the 1-dimensional signal is transformed into matrices for better capturing the dependencies.

**Strengths:**

**S1:** The research question raised by the paper is both valid and timely for the community.

**S2:** The paper is easy to follow, with graphs to show the model \& module designs, and visualised results for straightforward comparison and analysis.

**Weaknesses:**

**W1:** Inappropriate Averaging Across Horizons. Computing averages across horizons is misleading, as different forecasting horizons are different in forecasting difficulty, where averaging the errors could easily hide poor performance on some horizons.

**W2:** The claim (or the assumption) that there is "no inherent order among variables in multivariate time series" (line 290) does not always hold. For example, in weather forecasting, climate indicators at grid points closer to the target by nature have a greater influence than those farther away. Not considering this ordering would limit this work to applications where the covariates are iid, which is OK, as this can be the scope of the paper, but the authors need to make it clearer in the paper.

**W3:** Even if the covariates are iid, there is still concern that the reception field of the paper generally decides which variables are taken into modelling, this means that although the variables does not have an order themselves, the model performance would be highly dependent on the ordering as this influence which variables are modelled and which are not for each target variable.

**W4:** While the paper claims to have the dependency modelled dynamically over time, it is not clear how, just by looking at the modelling design. See Q3.

**W5:** Table 5 shows wildly different configurations per dataset, which raises concern that the good performance presented by the paper might be overfitting to benchmarks rather than a robust general solution.

**Questions:**

**Q1:** Modelling the dependencies dynamically over time has been proposed, either it is within the same channel [1] or across different channels [2,3], with the dynamics also captured with convolution layers and coarse and fine-grained dependencies captured at different temporal resolutions. Can the authors clarify what the key novelty of the proposed paper is compared to the prior work, and why the advocated way of capturing dynamic dependencies is better?

[1] DeformableTST: Transformer for Time Series Forecasting without Over-reliance on Patching (NeurIPS 2024)

[2] DeformTime: Capturing Variable Dependencies with Deformable Attention for Time Series Forecasting (TMLR 2025)

[3] Adaptive Convolutional Forecasting Network Based on Time Series Feature-Driven

**Q2:**
The results are averaged across the full forecasting sequence, i.e., [t+1,t+H]. However, this might over-credit the models that are giving better performance on time steps close to t+1, but bad performance when close to t+H, compared to models that do an average job across all time steps. How is FACT compared to baselines when only evaluated on the target forecasting horizon t+H?

**Q3:** Could the authors clarify the mechanism or provide more details on how the temporal dynamics of dependencies are captured?

---

> ### Author Response · Authors · 2025-11-24
>
> We appreciate your insightful comments and have addressed all of them below with additional experiments.
>
> ### **W1&Q2. How is FACT compared to baselines when only evaluated on the target forecasting horizon t+H?**
>
> **Response:** We first clarify that averaging errors across all forecasting horizons is the standard evaluation protocol in multivariate time series forecasting, as adopted by recent representative works such as PatchTST, iTransformer, and TimeFilter. We follow this setting to ensure fair comparison.
>
> We agree that evaluation at specific target horizons is also important. As suggested, we report performance at the target horizons (target time steps): t+3, t+6, t+12, and t+24. For a comprehensive comparison, we include several recent and competitive baselines: the latest attention-based models (iTransformer, TimeXer), the latest graph-based model (TimeFilter), and the latest Mamba-based model (TimePro). The results are summarized in **Table 1 below**, using MSE as the evaluation metric. Across all baselines and target horizons, FACT consistently demonstrates superior performance. Notably, as the forecasting horizon increases, the performance gap between FACT and other baselines generally widens. These results further validate the effectiveness of our proposed method under various evaluation settings.
>
> **Table 1. Performance comparison on the target forecasting horizon**
>
> |H|Model|PEMS03|PEMS04|PEMS08|METR-LA|Traffic2|PEMSD7|PEMS-BAY|
> |---|---|---|---|---|---|---|---|---|
> |3|FACT|**0.049**|**0.061**|**0.058**|**0.273**|**0.168**|**0.153**|**0.162**|
> ||TimeFilter|0.050|0.064|0.058|0.277|0.187|0.163|0.210|
> ||TimeXer|0.062|0.070|0.167|0.283|0.181|0.156|0.172||
> ||TimePro|0.058|0.069|0.161|0.313|0.183|0.163|0.218||
> ||iTransformer|0.063|0.069|0.161|0.326|0.188|0.163|0.175|
> ||
> |6|FACT|**0.059**|**0.068**|**0.069**|**0.438**|**0.184**|**0.249**|**0.270**||
> ||TimeFilter|0.063|0.075|0.070|0.443|0.206|0.284|0.350||
> ||TimeXer|0.073|0.078|0.186|0.439|0.196|0.253|0.276||
> ||TimePro|0.071|0.079|0.188|0.465|0.200|0.265|0.332||
> ||iTransformer|0.075|0.081|0.179|0.468|0.206|0.269|0.290|
> ||
> |12|FACT|**0.073**|**0.077**|**0.085**|**0.668**|**0.203**|**0.336**|**0.371**||
> ||TimeFilter|0.082|0.091|0.092|0.692|0.222|0.441|0.547||
> ||TimeXer|0.089|0.090|0.214|0.670|0.211|0.349|0.376||
> ||TimePro|0.092|0.097|0.240|0.686|0.218|0.375|0.455||
> ||iTransformer|0.098|0.102|0.218|0.677|0.221|0.387|0.413|
> ||
> |24|FACT|**0.098**|**0.097**|**0.117**|**0.927**|**0.222**|**0.414**|**0.451**||
> ||TimeFilter|0.129|0.131|0.146|1.014|0.232|0.661|0.875||
> ||TimeXer|0.124|0.115|0.268|0.943|0.227|0.427|0.478||
> ||TimePro|0.132|0.132|0.352|0.956|0.229|0.482|0.609||
> ||iTransformer|0.143|0.143|0.295|0.962|0.233|0.505|0.546|

---

> ### Author Response · Authors · 2025-11-24
>
> ### **W2. The claim (or the assumption) that there is "no inherent order among variables in multivariate time series" (line 290) does not always hold. Not considering this ordering would limit this work to applications where the covariates are iid, which is OK, as this can be the scope of the paper, but the authors need to make it clearer in the paper.**
> ### **W3. Even if the covariates are iid, there is still concern that the reception field of the paper generally decides which variables are taken into modelling, this means that although the variables does not have an order themselves, the model performance would be highly dependent on the ordering as this influence which variables are modelled and which are not for each target variable.**
>
> **Response:** For **W2**, our focus is on multivariate time series data, regardless of whether the variables have inherent order or not.  As you noted, it is indeed beyond the scope of this work to consider inherent order among variables. As suggested, we have revised the statement in the paper to make it clear that our method does not rely on any prior knowledge about the ordering among the $C$ variables in the input multivariate time series data.
>
> For **W3**, we have conducted experiments to demonstrate that our method is robust to variable permutations and various 2D reconstruction strategies. In our design, we do not assume any prior order among the $C$ variables; we simply reshape the input sequence of $C$ variables into an $H \times W$ grid without imposing additional ordering. Our method, FACT, performs the 2D reconstruction using row-major order by default and applies the proposed multi-dilated convolution architecture to ensure comprehensive consideration of all variables. Meanwhile, to assess the effect of variable order, we compare model performance when the variables are randomly shuffled. Additionally, to further demonstrate the robustness of our approach, we evaluate several alternative 2D reconstruction strategies, including column-major, Z-order, snake, spiral, and Hilbert curve layouts. The results summarized in **Table 2 below** show that FACT with different variable permutations exhibits only minor performance differences. This demonstrates the robustness of our approach to variable order and the generalizability of our method to various multivariate time series datasets, with or without inherent order.
>
>
> **Table 2. Comparison with different strategies on variable order**
>
> |Model|metric|PEMS03|PEMS04|PEMS08|METR-LA|Traffic2|PEMSD7|PEMS-BAY|
> |---|---|---|---|---|---|---|---|---|
> |FACT|MSE|0.086|0.087|0.107|0.704|0.204|0.330|0.372|
> ||MAE|0.194|0.194|0.200|0.456|0.179|0.316|0.267|
> |+random permutation|MSE|0.087|0.088|0.110|0.706|0.206|0.331|0.369
> ||MAE|0.196|0.196|0.202|0.458|0.180|0.336|0.270
> |+column-major|MSE|0.086|0.086|0.106|0.709|0.204|0.329|0.371|
> ||MAE|0.194|0.194|0.201|0.457|0.179|0.315|0.268|
> |+z-order|MSE|0.087|0.086|0.107|0.706|0.206|0.330|0.377|
> ||MAE|0.195|0.194|0.202|0.456|0.180|0.316|0.270|
> |+snake|MSE|0.087|0.087|0.104|0.707|0.205|0.330|0.369|
> ||MAE|0.194|0.195|0.199|0.456|0.179|0.316|0.267|
> |+spiral|MSE|0.086|0.087|0.108|0.705|0.206|0.332|0.375|
> ||MAE|0.195|0.195|0.201|0.454|0.179|0.317|0.270|
> |+hilbert|MSE|0.086|0.087|0.105|0.715|0.205|0.332|0.380|
> ||MAE|0.194|0.195|0.197|0.454|0.179|0.318|0.271|

---

> ### Author Response · Authors · 2025-11-24
>
> ### **W4&Q3. Clarify the dynamic modelling of dependencies over time**
> **Response:** We clarify that we have developed techniques to model both cross-variable interactions and temporal dependencies, as detailed in the paper. Specifically, we employ a stack of DConvBlocks (see Lines 204–205 and 226–235), each of which contains a Depth-wise Inception module to capture dynamic variable interactions and a feedforward network (FFN) to model temporal dependencies within each variable. In the paper, the term *dynamic* refers to the fact that variable interactions differ across different granularities. To capture these dynamic interactions at each granularity, we use the Depth-wise Inception module, which learns separate convolution kernels for different granularities. Meanwhile, as described in line 328 of the main text (Feed-Forward Networks section), our FFN module is designed to capture temporal dependencies within each variable. In the time domain, the FFN fuses information from different data points to model temporal dependencies, thereby enhancing the representation. In the frequency domain, the FFN aggregates information across frequency components within each variable, capturing phenomena such as amplitude coupling, phase coupling, and cross-frequency modulation to model temporal dependencies in the frequency space.
>
>
>
> ### **W5. Parameter Configurations.**
> **Response:** First, we clarify that it is a well-established practice in the literature of multivariate time series forecasting to tune certain hyperparameters and provide the chosen configuration as we reported in the paper.  The batch size, learning rate, the number of kernels $k$ are all the same across all twelve datasets. We have set the dimension $D$ 512, the same for most datasets. Moreover, we have conducted experiments to vary the parameters to study their impact on performance. We find that the model is generally robust to change within a reasonable range.  For example, in Figure 6 of the Appendix, we have varied $\alpha$ and in Figure 7 of the Appendix, we have varied the number of kernels $k$.  To analyze the sensitivity to the number of layers, we conduct comparison experiments on four datasets with different numbers of variables (consistent with those used in the main text). We evaluate several layer configurations: 3 layers ($n=[1,2,1]$), 4 layers ($n=[1,2,2,1]$), 5 layers ($n=[1,2,3,2,1]$), 6 layers ($n=[1,2,3,3,2,1]$), and 7 layers ($n=[1,2,3,4,3,2,1]$). From the 3-layer to the 7-layer configuration, the receptive field expands progressively, enabling the model to capture increasingly broader dependency ranges.The results are summarized **in Table 3 below**.  Increasing the number of layers enlarges the receptive field, and overall, performance tends to improve or remain stable. Different datasets exhibit varying sensitivities to receptive field size due to their distinct characteristics. On most datasets, such as Traffic2, PEMS04, and PEMS08, expanding the receptive field by increasing the number of layers leads to stable or slightly improved performance, while on METR-LA, further increasing the receptive field beyond a certain point results in performance degradation. Therefore, we set the default number of layers to 5.
>
> **Table 3. Sensitivity analysis of layers**
> |Model|metric|Traffic2|PEMS04|PEMS08|METR-LA
> |---|---|---|---|---|---|
> |3 layers|MSE|0.210|0.091|0.112|0.704
> ||MAE|0.182|0.201|0.207|0.456
> |4 layers|MSE|0.207|0.088|0.107|0.705
> ||MAE|0.180|0.196|0.202|0.458
> |5 layers|MSE|0.204|0.086|0.107|0.711
> ||MAE|0.179|0.194|0.194|0.459
> |6 layers|MSE|0.202|0.086|0.107|0.715
> ||MAE|0.179|0.194|0.198|0.460
> |7 layers|MSE|0.201|0.085|0.109|0.729
> ||MAE|0.178|0.192|0.200|0.463|

---

> ### Author Response · Authors · 2025-11-24
>
> ### **Q1. Clarify what the key novelty of the proposed paper is compared to the prior work, and why the advocated way of capturing dynamic dependencies is better**
>
> **Response:** As suggested, we further clarify the key novelty of our work compared to prior methods and highlight our technical advantages. As discussed in the introduction, complex variable interactions in multivariate time series often vary across different granularities. Capturing these dynamic interactions enables the model to better understand how variables influence one another, thereby improving forecasting performance. However, most existing methods primarily capture coarse-grained correlations. To address this, we propose FACT, which explicitly models fine-grained variable interactions from both the time and frequency domains at multiple granularities. This approach differs from DeformableTST [1], which uses a deformable attention mechanism to primarily model temporal dependencies and treats variables independently. FACT also differs from DeformTime [2] and ACNet [3], which mainly capture relationships among adjacent variables in the time domain through learned positional offsets and focus on local temporal patterns, rather than modeling all variable interactions at each granularity.
>
> FACT introduces a depth-wise convolution block (DConvBlock) that employs depth-wise convolutions with specific kernels to explicitly model variable interactions at each granularity. FACT utilizes a stack of DConvBlocks to progressively capture interactions. In contrast, DeformTime uses patching and attention mechanisms to model interactions of neighboring variables through learned offsets in key and value within temporal patches, and ACNet captures local nonlinear variable features through gated deformable convolution for local nonlinear temporal patterns, rather than at each granularity. Furthermore, we reshape the 1D variable into 2D space and design multi-dilated convolution kernels with progressively increasing dilation rates. This design, with relatively few parameters, achieves a broader receptive field to effectively capture interactions among all variables. In contrast, DeformTime models interactions only between neighboring variables and incurs high computational cost (as reported in Appendix E.5 of the DeformTime paper), while ACNet employs deformable convolution with fixed kernel size, resulting in a limited receptive field that cannot expand to global interactions. Extensive experiments on multiple real-world datasets demonstrate the effectiveness of our proposed method.
>
> [1] DeformableTST: Transformer for Time Series Forecasting without Over-reliance on Patching (NeurIPS 2024)
>
> [2] DeformTime: Capturing Variable Dependencies with Deformable Attention for Time Series Forecasting (TMLR 2025)
>
> [3] Adaptive Convolutional Forecasting Network Based on Time Series Feature-Driven

---

> ### Comment · Reviewer_5JYU · 2025-11-25
> **Rebuttal acknowledged**
>
> Thank the authors for the detailed clarification and experiments. I've raised my score from 4 to 6.

---

> > ### Author Response · Authors · 2025-11-25
> > **Thank you**
> >
> > Thank you for raising the score. We also appreciate your efforts.

---

### Official Review · Reviewer_4AVh · 2025-10-30

**Soundness:** 3
**Presentation:** 3
**Contribution:** 2
**Rating:** 6
**Confidence:** 4

**Summary:**

This paper proposes the FACT architecture for multivariate time series forecasting (MTSF), designed to capture fine-grained dynamic interactions between variables in both the time and frequency domains. Its core component, the DConvBlock, uses depth-wise convolution to model variable relationships individually at each time step or frequency component (termed 'granularity' in the paper). In the frequency domain, it notably processes the signal by separating it into Amplitude (A) and Phase (P), modeling them independently. Furthermore, it restructures the 1D variable dimension into a 2D space and applies multi-dilated 2D convolution, efficiently achieving a global receptive field across all variables with fewer layers. Experimental results show that FACT achieves SOTA accuracy on 12 benchmarks and significantly reduces training time and memory consumption compared to attention mechanisms.

**Strengths:**

1. Fine-grained Interaction Modeling: Unlike existing models that focus on macro-level relationships of the entire variable set, the idea of capturing dynamic interactions at a fine-grained 'granularity' level (i.e., 'each time step' and 'each frequency component') is highly compelling. The method of assigning independent kernels to each channel (granularity) using depth-wise convolution to achieve this is technically novel and effective.
2. Robust Dual-Domain Design: The dual-domain architecture, which considers both instantaneous interactions in the time domain and periodic resonances/phase relationships in the frequency domain, is very robust. The approach of modeling the frequency domain by separating it into Amplitude and Phase, which allows for more direct physical interpretation rather than just Real/Imaginary parts, is impressive. The ablation study (Table 3) clearly demonstrates that these two domains are complementary.
3. Modularity and Generalizability: The core module, MDInception (the convolutional part of DConvBlock), was shown to improve performance not only within FACT but also when integrated into other baseline models (TimesNet, PatchTST, iTransformer) (Appendix E.1). Notably, it significantly boosted the performance of the variable-independent model PatchTST. This suggests the proposed module can function as a general-purpose solution for capturing inter-variable interactions.

**Weaknesses:**

1. Limitation of the 2D Reshaping Assumption: The paper assumes "no inherent order among variables" and restructures the 1D variable list into a 2D grid. However, traffic datasets like PEMS and METR-LA possess an underlying spatial structure (road networks), where the variable (sensor) ID order might reflect adjacency information. This reshaping method, which ignores potential spatial locality, could destroy critical information. There is insufficient discussion on whether this assumption holds true for all 12 datasets.
2. Lack of Interpretability for 'Granularity-Specific Learning': This is a core claim of the paper. However, there is no qualitative analysis (e.g., visualization) to verify what the depth-wise convolution kernels actually learn. For example, it is not shown whether specific kernels (channels) in the frequency-domain DConvBlock truly specialize in 24-hour periodicity versus 7-day periodicity, or whether time-domain kernels learn to capture interactions between specific variable pairs (like in Fig 1b) or specific time lags.
3. Questionable Efficacy of A/P Transformation: In Appendix D (Table 6), the paper compares the proposed Amplitude/Phase (A/P) separation modeling with the Real/Imaginary (R/I) alternative. However, the results from both methods are nearly identical (e.g., PEMS03 MSE 0.194 vs 0.193, METR-LA MSE 0.704 vs 0.706). The A/P method requires additional non-linear operations (arctan, cos, sin), whereas R/I is a simple separation. Contrary to the paper's claim of being "consistently the best," the data does not show a clear advantage for the A/P method over the R/I method. This weakens the justification for choosing the more complex A/P representation.

**Questions:**

1. You assume no inherent order among variables for 2D reshaping. For datasets like PEMS or METR-LA where actual spatial adjacency between sensors is critical, did you verify if the mapping method (e.g., row-major, z-order curve) from 1D variable IDs to the 2D grid impacts performance?
2. To support the core claim of 'granularity-specific learning,' could you provide visualizations (e.g., kernel weights, activation maps) to demonstrate that the learned DConvBlock kernels actually capture specific frequency bands or dynamic relationships between variable pairs?
3. In Appendix D (Table 6), the performance advantage of the amplitude/phase separation method over real/imaginary or complex convolution methods is not significant. Beyond 'physical interpretation,' were there any practical advantages (e.g., faster computation, better training stability) to adopting the amplitude/phase method over the simpler real/imaginary method?
4. The paper uses a 5-layer DConvBlock. Is there a specific justification or ablation study for setting the number of layers (N) to 5? A sensitivity analysis of performance and receptive field changes based on the number of layers seems necessary.
5. You used the Inception module as the convolution backbone, but Appendix C shows that standard 2D convolution ('+Conv') also achieves competitive performance. How does the multi-branch structure of Inception compare to standard 2D convolution in terms of parameter count or FLOPs? Is the performance increase sufficient to justify the additional cost?
6. You used a modified Inception as the backbone for DConvBlock. Did you consider combining the depth-wise convolution with a mechanism that explicitly models relationships between channels (i.e., 'granularity' in this paper), such as a Squeeze-and-Excite (SENet) block?

---

> ### Author Response · Authors · 2025-11-24
>
> Thank you for your constructive comments. We have addressed all your comments below with additional experiments.
>
> ### **W1&Q1. The paper assumes "no inherent order among variables" and restructures the 1D variable list into a 2D grid.   You assume no inherent order among variables for 2D reshaping. For datasets like PEMS or METR-LA where actual spatial adjacency between sensors is critical, did you verify if the mapping method (e.g., row-major, z-order curve) from 1D variable IDs to the 2D grid impacts performance?**
>
> **Response:** We clarify that our work focuses on multivariate time series data, and incorporating additional information such as spatial adjacency is beyond the scope of this paper. Our method does not utilize prior knowledge about spatial adjacency or ordering among the $C$ variables in the input data, even if such information is available. Therefore, we do not assume an "inherent order among variables" (we have revised the paper to clarify this point); our method simply takes the input sequence of $C$ variables and reshapes it into an $H \times W$ grid using row-major order.
>
> In our experiments, we evaluate only on multivariate time series datasets and do not utilize spatial adjacency information, even for datasets like PEMS or METR-LA where such information is available.
>
> Our method, FACT, reshapes the input sequence of $C$ variables into an $H \times W$ grid using row-major order, where the first $W$ variables are placed in the first row, the next $W$ in the second row, and so on. As suggested, we compare with additional mapping methods from the 1D variable list to a 2D grid, including: (1) random permutation of all variables, and (2) various 1D-to-2D layout mappings, such as column-major, z-order, snake, spiral, and Hilbert curve. The results (**summarized in Table 1 below**) show that performance variations across different layouts are marginal, demonstrating that our model is stable and robust with respect to variable order. This confirms that our method does not rely on any specific variable arrangement and remains robust under different variable arrangements, validating the generalizability of our approach to various multivariate time series datasets with or without inherent spatial adjacency.
>
> **Table 1. Comparison of different 1D-to-2D mappings**
>
> |Model|metric|PEMS03|PEMS04|PEMS08|METR-LA|Traffic2|PEMSD7|PEMS-BAY|
> |---|---|---|---|---|---|---|---|---|
> |FACT|MSE|0.086|0.087|0.107|0.704|0.204|0.330|0.372|
> ||MAE|0.194|0.194|0.200|0.456|0.179|0.316|0.267|
> ||
> |+random permutation|MSE|0.087|0.088|0.110|0.706|0.206|0.331|0.369
> ||MAE|0.196|0.196|0.202|0.458|0.180|0.336|0.270
> ||
> |+column-major|MSE|0.086|0.086|0.106|0.709|0.204|0.329|0.371|
> ||MAE|0.194|0.194|0.201|0.457|0.179|0.315|0.268|
> |+z-order|MSE|0.087|0.086|0.107|0.706|0.206|0.330|0.377|
> ||MAE|0.195|0.194|0.202|0.456|0.180|0.316|0.270|
> |+snake|MSE|0.087|0.087|0.104|0.707|0.205|0.330|0.369|
> ||MAE|0.194|0.195|0.199|0.456|0.179|0.316|0.267|
> |+spiral|MSE|0.086|0.087|0.108|0.705|0.206|0.332|0.375|
> ||MAE|0.195|0.195|0.201|0.454|0.179|0.317|0.270|
> |+hilbert|MSE|0.086|0.087|0.105|0.715|0.205|0.332|0.380|
> ||MAE|0.194|0.195|0.197|0.454|0.179|0.318|0.271|

---

> ### Author Response · Authors · 2025-11-24
>
> ### **W2&Q2. Visualization of Granularity-Specific Learning**
>
> **Response:** As suggested, we have enhanced the interpretability of our model by providing visualization of variable correlations and activation maps at different granularities.
>
> Specifically, we visualize the overall correlation matrix of all variables in the time domain together with the corresponding activation map of our time-domain convolution, and the variable amplitude-correlation and phase-correlation matrices in the frequency domain together with the activation maps of amplitude and phase interactions from our frequency-domain convolution. Results for three different datasets are reported in **Figures 16(a), 17(a), and 18(a) in the revised paper**.
> From these visualizations, we observe that variable pairs with strong correlations exhibit strong responses in the learned activation maps, whereas weakly correlated pairs show weak activation responses. These results indicate that our model effectively captures cross-variable interactions and provide direct supporting evidence for the interpretability and reliability of our variable interaction modeling.
>
>
> Additionally, we also visualize variable correlation and activation maps across multiple granularities, including data points in the time domain and frequencies in the frequency domain. At each granularity, we present the variable correlation matrix in the time domain alongside the corresponding activation map, and the amplitude-correlation and phase-correlation matrices in the frequency domain together with activation maps capturing amplitude and phase interactions. For clearer illustration, we include three different datasets and randomly select ten variables for visualization. The results are reported in **Figures 16(b), 17(b), and 18(b) in the revised paper**.
> Our findings show that, for the same dataset, variable correlations can differ substantially across granularities, which aligns with our model design that explicitly captures dynamic variable interactions at each granularity. Moreover, within each granularity, strongly correlated variables consistently exhibit higher activation responses, while weakly correlated variables show weaker responses, both in the time and frequency domains. This consistent pattern demonstrates that our convolution structure effectively captures fine-grained cross-variable interactions.
>
>
> ### **W3&Q3. Efficacy of A/P Transformation**
> **Response:** Thank you for this comment. We clarify that our method supports frequency-domain modeling using either the Amplitude/Phase (A/P) transformation or the Real/Imaginary (R/I) alternative. Empirically, we find that A/P generally yields strong performance, though the differences are marginal in some cases. We have revised the relevant statements in the paper to reflect this observation accurately.
>
> To further assess the practical benefits of the A/P transformation over the R/I alternative, we conducted a convergence analysis under identical parameter settings. While the A/P transformation introduces additional non-linear operations (arctan, cos, sin), it does not increase the number of trainable parameters. We evaluated both methods on four datasets and visualized their test performance throughout training. As shown in **Figure 11 of the revised paper**, two limitations of the R/I alternative are evident: (1) it often fails to converge to the global optimum, negatively impacting final performance; and (2) even when near-optimal solutions are reached, R/I exhibits greater instability and larger performance fluctuations compared to A/P. Note that the use of A/P is not a central design choice, but rather one option for frequency-domain modeling. The main contribution of our method lies in the overall architecture, specifically the depth-wise multi-dilation convolution and 2D variable reconstruction techniques, which are orthogonal to the choice of frequency representation. Therefore, we adopt the A/P transformation in practice.

---

> > ### Author Response · Authors · 2025-11-24
> >
> > ### **Q4. Sensitivity analysis about the number of layer.**
> > **Response:** As suggested, we conduct a sensitivity analysis by varying the number of layers and the corresponding dilation rates $n$, which are directly related to the number of layers. Specifically, we evaluate several configurations: 3 layers ($n=[1,2,1]$), 4 layers ($n=[1,2,2,1]$), 5 layers ($n=[1,2,3,2,1]$), 6 layers ($n=[1,2,3,3,2,1]$), and 7 layers ($n=[1,2,3,4,3,2,1]$). As the number of layers increases from 3 to 7, the receptive field expands, enabling the model to capture broader dependency ranges. In our model, we employ a 5-layer DConvBlock with default multi-dilation rates of [1,2,3,2,1]. The results are summarized **in Table 2 below**. Increasing the number of layers enlarges the receptive field, and overall, performance tends to improve or remain stable. Different datasets exhibit varying sensitivities to receptive field size due to their distinct characteristics. On most datasets, such as Traffic2, PEMS04, and PEMS08, expanding the receptive field by increasing the number of layers leads to stable or slightly improved performance, while on METR-LA, further increasing the receptive field beyond a certain point results in performance degradation. Therefore, we set the default number of layers to 5.
> >
> >
> > **Table 2. Sensitivity analysis of layers**
> > |Model|metric|Traffic2|PEMS04|PEMS08|METR-LA|
> > |---|---|---|---|---|---|
> > |3 layers|MSE|0.210|0.091|0.112|0.704|
> > ||MAE|0.182|0.201|0.207|0.456|
> > |4 layers|MSE|0.207|0.088|0.107|0.705|
> > ||MAE|0.180|0.196|0.202|0.458|
> > |5 layers|MSE|0.204|0.086|0.107|0.711|
> > ||MAE|0.179|0.194|0.194|0.459|
> > |6 layers|MSE|0.202|0.086|0.107|0.715|
> > ||MAE|0.179|0.194|0.198|0.460|
> > |7 layers|MSE|0.201|0.085|0.109|0.729|
> > ||MAE|0.178|0.192|0.200|0.463|
> >
> > ### **Q5. Parameter count and FLOPs between Inception and Conv**
> >
> > **Response:** As suggested, we compare our method FACT with the Inception module against a variant where the Inception module is replaced by a standard 2D convolution. We report effectiveness measured by MSE, along with parameter count and FLOPs in **Table 3 below**. The results demonstrate that FACT with the Inception module achieves better effectiveness than the standard 2D convolution, while introducing only a marginal increase in parameters and FLOPs. Since each kernel at each channel has only $1 \times H \times W$ parameters, the Inception module increases the number of kernel elements only slightly compared to a standard 2D convolution. This represents a worthwhile trade-off for improved effectiveness.
> >
> > **Table 3. parameter and FLOPs comparison**
> >
> > |Model|metric|PEMS03|PEMS08|METR-LA|ECL|Traffic|Solar
> > |---|---|---|---|---|---|---|---|
> > |FACT|parameter (M)|21.94|11.07|3.63|31.05|21.98|3.63
> > ||FLOPs (G)|505.98|138.34|52.08|715.74|1263.74|33.34
> > ||MSE|0.086|0.107|0.704|0.157|0.425|0.196
> > |+Conv|parameter (M)|21.59|10.89|3.47|30.43|21.63|3.47
> > ||FLOPs (G)|498.52|136.32|50.02|702.46|1225.12|32.02
> > ||MSE|0.087|0.108|0.711|0.160|0.431|0.199
> >
> >
> > ### **Q6. You used a modified Inception as the backbone for DConvBlock. Did you consider combining the depth-wise convolution with a mechanism that explicitly models relationships between channels (i.e., 'granularity' in this paper), such as a Squeeze-and-Excite (SENet) block?**
> >
> > **Response:** Thank you for suggesting this technique. As recommended, we evaluated our method FACT with its variant incorporating the SENet block. The results, summarized in **Table 4 below**, show that FACT with and without the SENet block achieve comparable performance across all datasets. We attribute this to our point-wise FFN module following the depth-wise Inception block, which already effectively models relationships between channels. Consequently, adding the SENet block does not yield additional performance gains.
> >
> >
> > **Table 4. Comparison of combining SENet block**
> >
> > |Model|metric|PEMS03|PEMS04|PEMS08|METR-LA|Traffic2|ECL|Traffic|Solar|PEMSD7|PEMS-BAY|
> > |---|---|---|---|---|---|---|---|---|---|---|---|
> > |FACT|MSE|0.086|0.087|0.107|0.704|0.204|0.157|0.425|0.196|0.330|0.372|
> > ||MAE|0.194|0.194|0.200|0.456|0.179|0.254|0.276|0.254|0.316|0.267|
> > |+SENet|MSE|0.088|0.086|0.106|0.718|0.201|0.158|0.430|0.199|0.330|0.371
> > ||MAE|0.196|0.194|0.202|0.473|0.178|0.255|0.277|0.261|0.316|0.270

---

> > > ### Comment · Reviewer_4AVh · 2025-11-27
> > >
> > > I appreciate the thoughtful and well-structured response. The additional analyses and clarifications were helpful in addressing my concerns. I've updated my score accordingly.

---

> > > > ### Author Response · Authors · 2025-11-27
> > > > **Thank you**
> > > >
> > > > Thank you for raising the score. We also appreciate your efforts.

---

### Official Review · Reviewer_9MyL · 2025-10-31

**Soundness:** 2
**Presentation:** 3
**Contribution:** 3
**Rating:** 6
**Confidence:** 4

**Summary:**

This paper proposes FACT, a novel convolutional architecture for multivariate time-series forecasting. The model aims to capture fine-grained, time-varying interactions between variables. Its core contributions include: 1) a depth-wise convolution module named DConvBlock to capture dynamic interactions; 2) a strategy of reshaping the 1D variable list into a 2D grid to leverage efficient 2D convolutions; and 3) a dual-domain modeling approach combining time and frequency domains. The authors conduct extensive experiments on 12 benchmark datasets, claiming state-of-the-art (SOTA) performance in both prediction accuracy and computational efficiency.

On the positive side: Empirically, it achieves SOTA results on 12 benchmarks, which is a strong contribution. Furthermore, its significant improvement in computational efficiency (compared to the $O(N^2)$ complexity of Transformers) holds high practical value and appeal.
On the debatable side: The theoretical depth of its core methodology (2D variable reconstruction) needs strengthening. The long-term contribution and guiding effect of this strategy—which trades physical priors for efficiency—on the spatio-temporal data mining field are worthy of further discussion.

**Strengths:**

1.SOTA Empirical Performance: The most significant strength is that the paper comprehensively outperforms 12 recent models, including iTransformer, across 12 standard benchmarks. This is a very solid engineering achievement.
2.High Computational Efficiency: Compared to the $O(N^2)$ complexity of attention-based models (where N is the number of variables), the proposed convolutional architecture (especially after 2D reconstruction) offers a substantial advantage for high-dimensional variables. The reported reductions in training time and memory consumption (up to 50%) are highly attractive practical features.
3.Clever Design of DConvBlock: The use of depth-wise convolution to assign a dedicated kernel for each time/frequency granularity, as a means to capture "dynamic interactions," is a lightweight and effective engineering design.
4.Thorough Ablation Studies: The authors have conducted detailed ablation studies that demonstrate the necessity of the model's various components (e.g., dual-domain modeling, multi-dilated convolution).

**Weaknesses:**

1.Questionable Rationale of 2D Variable Reconstruction: A primary concern is the "2D variable reconstruction" strategy. This index-based reshaping appears to lack prior support. This operation could potentially place physically unrelated variables adjacent to each other while distancing physically adjacent ones. This approach may not fully leverage the valuable physical-topological priors inherent in spatio-temporal data.

2.Omission of GNN Baselines: This is a noticeable omission. The paper claims SOTA on several datasets with clear spatio-temporal structures (e.g., PEMS, METR-LA, Traffic) but fails to compare against Graph Neural Network (GNN) models specifically designed for such data (e.g., DCRNN, Graph WaveNet, MTGNN). GNNs are a standard baseline in spatio-temporal forecasting, and lacking this comparison somewhat weakens the convincingness of the SOTA claims on these specific datasets.

3.Insufficient Theoretical Discussion: As mentioned, the paper does not provide adequate theoretical justification for the "2D variable reconstruction." It relies heavily on empirical results for validation but lacks a deeper theoretical exploration of why this artificial pseudo-topology is effective, which is a slight pity for an ICLR paper.

4.Interpretability Needs Improvement: As the 2D grid is an artificially constructed "pseudo-space," the weights of the 2D convolution kernels may be difficult to map directly to real-world physical interactions, posing a challenge to the model's interpretability.

**Questions:**

1.Could the authors further elaborate on the rationale for the "2D variable reconstruction" from a theoretical perspective (beyond just "it works empirically")? Why would an arbitrary, index-based grid be a better representation for variable relationships than the original 1D list or a true physical graph (as used by GNNs)?

2.Why were GNN models omitted from the baseline comparison? Given that PEMS and METR-LA are standard GNN benchmarks, we strongly suggest the authors add comparisons against mainstream GNN models to truly substantiate the SOTA claim.
3.Have the authors experimented with other reshaping methods, such as clustering-based reshaping, or projecting the sensors' true physical coordinates (latitude/longitude) onto a 2D space and then applying 2D convolution on this physically meaningful 2D image? This would at least preserve spatial locality.

4.Permutation Sensitivity: The 2D reconstruction strategy seems dependent on the original index order of variables in the dataset. If the order of variables is permuted (e.g., swapping variable 5 and variable 50), would the model's performance change significantly?

5.Generality of GNNs vs. Necessity of Topology: The authors mention GNNs in the related work but seem to avoid them in comparisons. Do the authors perceive GNN models (which require an adjacency matrix) as lacking "generality," whereas FACT (which does not) is more general? If so, does FACT's SOTA performance on these spatio-temporal datasets imply a deeper conclusion: that physical-topological priors are not necessary for forecasting in these specific benchmarks, or perhaps that their importance has been overestimated?

6.Physical Interpretation of 2D Reshaping in Frequency Domain: For frequency domain modeling, the variable dimension C is similarly reshaped into $\sqrt{C} \times \sqrt{C}$. This operation seems even more abstract. In the time domain, "variable 1" and "variable 11" becoming neighbors is debatable; in the frequency domain, what is the physical meaning of applying a 2D convolution to the "amplitude/phase of variable 1" and the "amplitude/phase of variable 11"? How does the model learn meaningful patterns in such a highly abstract 2D frequency space?

---

> ### Author Response · Authors · 2025-11-24
>
> We appreciate your valuable feedback and have addressed all your comments below with additional experiments.
>
> ### **W1&Q3. About 2D Variable Reconstruction; Comparison with other reshaping methods, such as clustering-based reshaping**
>
> **Response:**   First, we clarify that we do not assume any prior knowledge about the order among the $C$ variables, such as physical adjacency, when reshaping them into an $H \times W$ grid for 2D variable reconstruction. Our work focuses on time series data, regardless of whether the variables have inherent physical order or not. Our design ensures broad applicability to multivariate time series forecasting tasks, without requiring spatial information or imposing constraints based on variable order. Therefore, when consutrcting the 2D grid, we simply reshape the $C$ variables into an $H \times W$ grid using row-major order, taking the first $W$ variables in the first row, the next $W$ in the second row, and so on, following the input sequence.
>
> Second, our method is agnostic to specific reshaping strategies and can flexibly accommodate various 2D layouts. As suggested, we conducted experiments to evaluate the impact of incorporating physical topology by comparing with (1) Clustering-based layout, where variables are grouped by correlation strength in 1D and then reshaped into a 2D grid, positioning highly correlated variables closer together; (2) MDS-based layout, where Multidimensional Scaling (MDS) projects variables into 2D space, preserving correlation-based proximity. While clustering-based layouts partially capture topology, MDS-based layouts offer a more rigorous representation. The results in **Table 1 below** show that the performance of these two variant layouts is comparable to our original FACT model. This demonstrates that our model does not rely on physical topology for effective forecasting, and thus is generalizable to various datasets without inherent variable order, which is a key advantage of our approach.
>
> **Table 1. Comparison with introducing physical topology**
>
> |Model|metric|PEMS03|PEMS04|PEMS08|METR-LA|Traffic2|PEMSD7|PEMS-BAY|
> |---|---|---|---|---|---|---|---|---|
> |FACT|MSE|0.086|0.087|0.107|0.704|0.204|0.330|0.372|
> ||MAE|0.194|0.194|0.200|0.456|0.179|0.316|0.267|
> |+cluster|MSE|0.087|0.085|0.113|0.717|0.204|0.328|0.370|
> ||MAE|0.195|0.191|0.204|0.458|0.179|0.315|0.271|
> |+MDS|MSE|0.086|0.085|0.106|0.699|0.203|0.332|0.371|
> ||MAE|0.194|0.192|0.198|0.457|0.179|0.315|0.271|
>
>
>
> ### **Q4. The 2D reconstruction strategy seems dependent on the original index order of variables in the dataset. If the order of variables is permuted (e.g., swapping variable 5 and variable 50), would the model's performance change significantly?**
>
> **Response:** As clarified in our response to W1&Q3 above, our method does not assume any prior order among the $C$ variables and is designed for time series data, regardless of whether the variables have inherent physical order or not. Thus, we just take what is given in the input sequence and reshape the input sequence of $C$ variables into an $H \times W$ grid without imposing any additional ordering. To assess the effect of variable permutation, we compare model performance when the variables are randomly shuffled (e.g., swapping variable 5 and variable 50). Additionally, to further demonstrate the robustness of our approach, we evaluate several alternative 2D reconstruction strategies, including column-major, Z-order, snake, spiral, and Hilbert curve layouts. The results (summarized in **Table 2 below**) show that random permutations of variable order have minimal impact on model performance, indicating that our method remains stable and effective even when variable indices are swapped. Similarly, performance differences across various 2D reconstruction strategies are negligible, further validating that our approach does not rely on specific variable adjacency or arrangement for accurate forecasting.
>
>
> **Table 2. Comparison of different 2D reconstruction strategies**
>
> |Model|metric|PEMS03|PEMS04|PEMS08|METR-LA|Traffic2|PEMSD7|PEMS-BAY|
> |---|---|---|---|---|---|---|---|---|
> |FACT|MSE|0.086|0.087|0.107|0.704|0.204|0.330|0.372|
> ||MAE|0.194|0.194|0.200|0.456|0.179|0.316|0.267|
> ||
> |+random permutation|MSE|0.087|0.088|0.110|0.706|0.206|0.331|0.369
> ||MAE|0.196|0.196|0.202|0.458|0.180|0.336|0.270
> ||
> |+column-major|MSE|0.086|0.086|0.106|0.709|0.204|0.329|0.371|
> ||MAE|0.194|0.194|0.201|0.457|0.179|0.315|0.268|
> |+z-order|MSE|0.087|0.086|0.107|0.706|0.206|0.330|0.377|
> ||MAE|0.195|0.194|0.202|0.456|0.180|0.316|0.270|
> |+snake|MSE|0.087|0.087|0.104|0.707|0.205|0.330|0.369|
> ||MAE|0.194|0.195|0.199|0.456|0.179|0.316|0.267|
> |+spiral|MSE|0.086|0.087|0.108|0.705|0.206|0.332|0.375|
> ||MAE|0.195|0.195|0.201|0.454|0.179|0.317|0.270|
> |+hilbert|MSE|0.086|0.087|0.105|0.715|0.205|0.332|0.380|
> ||MAE|0.194|0.195|0.197|0.454|0.179|0.318|0.271|

---

> ### Author Response · Authors · 2025-11-24
>
> ### **W2&Q2&Q5. Comparison with GNN baselines.**
> **Response:** As suggested, we have conducted experiments to compare with several representative GNN models: MTGNN, DCRNN, AGCRN, Graph WaveNet, STGCN, and the latest TimeFilter. As shown in **Table 3** below. We can note from the results that our method achieves superior performance compared to all these GNN baselines, including the latest GNN model TimeFilter. These results indicate that our approach is effective even when compared against latest GNN models, and it validates the effectiveness of our proposed techniques in the paper. Moreover, we did not say that GNNs lack generality; rather, our model FACT adopts a different design methodology achieves fine-grained across-variable convolution, and our method is designed for multivariate time series forecasting, regardless of whether the input data has inherent physical order or not.
>
> **Table 3. Comparison with GNN baselines**
>
> |Model|metric|PEMS03|PEMS04|PEMS07|PEMS08|METR-LA|PEMSD7|PEMS-BAY|
> |---|---|---|---|---|---|---|---|---|
> |FACT|MSE|**0.086**|**0.087**|**0.073**|**0.107**|**0.704**|**0.330**|**0.372**|
> ||MAE|**0.194**|**0.194**|**0.170**|**0.200**|**0.456**|**0.316**|**0.267**|
> |DCRNN|MSE|0.273|0.243|0.479|0.542|0.915|0.644|0.704|
> ||MAE|0.380|0.360|0.358|0.426|0.602|0.529|0.479||
> |GraphWaveNet|MSE|0.186|0.184|0.159|0.261|0.884|0.443|0.499|
> ||MAE|0.289|0.290|0.266|0.304|0.551|0.392|0.345|
> |MTGNN|MSE|0.198|0.195|0.174|0.294|0.895|0.532|0.609||
> ||MAE|0.298|0.302|0.281|0.327|0.577|0.439|0.393||
> |AGCRN|MSE|0.157|0.150|0.177|0.266|0.748|0.442|0.510||
> ||MAE|0.269|0.267|0.275|0.292|0.501|0.372|0.360||
> |STGCN|MSE|0.101|0.092|0.094|0.208|0.732|0.343|0.396||
> ||MAE|0.208|0.203|0.182|0.240|0.473|0.321|0.291||
> |TimeFilter|MSE|0.108|0.114|0.101|0.139|0.819|0.498|0.547||
> ||MAE|0.217|0.226|0.207|0.230|0.493|0.404|0.358||
>
> ### **W3&Q1. Further discuss the rationale of 2D variable reconstruction**
>
> **Response:** Building on our responses to W1&Q3 and Q4 above, our method is designed for multivariate time series data, which often do not necessarily have inherent physical order among variables. Consequently, our 2D variable reconstruction technique does not depend on any specific physical topology. We simply reshape the input sequence of $C$ variables into an $H \times W$ grid, taking the input order without imposing additional requirement.
>
> Importantly, the 2D convolution grid is complemented by our multi-dilated convolution architecture, as described in Section 3.2 Multi-dilated architecture (starting from Line 297 and Figure 3 in the paper). By applying multiple dilation rates, we significantly expand the receptive field, enabling the model to efficiently capture variable interactions and provide dense, flexible coverage of the input space at each scale. This design, together with the depth-wise structure, achieves granularity-aware modeling and efficient receptive field expansion. As a result, our method effectively captures complex interactions among all $C$ variables, regardless of their arrangement on the 2D grid. The 2D reconstruction facilitates convolutional modeling, while the multi-dilated architecture ensures comprehensive consideration of all variables without relying on physical adjacency. As shown in Figure 10 in the Appendix, our method outperforms 1D convolutions under matched parameters and model layers, validating the effectiveness of our 2D reconstruction and multi-dilated architecture.

---

> ### Author Response · Authors · 2025-11-24
>
> ### **W4. Interpretability Needs Improvement: As the 2D grid is an artificially constructed "pseudo-space," the weights of the 2D convolution kernels may be difficult to map directly to real-world physical interactions, posing a challenge to the model's interpretability.**
>
> **Response:** To improve the interpretability as suggested, we provide a visualization analysis. Specifically, we visualize the overall correlation matrix of all variables in the time domain alongside the corresponding activation map of our time-domain convolution, as well as the variable amplitude-correlation matrix and variable phase-correlation matrix in the frequency domain together with the activation maps of amplitude and phase interactions from our frequency-domain convolution. We include results on three different datasets, as reported in **Figures 16(a), 17(a), and 18(a) in the revised paper**. These visualizations show that variable pairs with strong correlations exhibit strong responses in the learned activation maps, while weakly correlated pairs show weak activation responses. This demonstrates that our model effectively captures cross-variable interactions, even though the 2D grid is not tied to a true physical topology. These results provide direct evidence for the interpretability and reliability of our variable interaction modeling, and further show that our model can perform multivariate time series forecasting on real-world datasets without requiring additional information about physical topology among variables.
>
>
> ### **Q6. Physical Interpretation of 2D Reshaping in Frequency Domain**
>
> **Response:** We clarify that we reshape the 1D variable sequence into a 2D grid, but *do not* reshape along the amplitude/phase per variable in the frequency domain. Each variable in the 2D convolution grid retains its own amplitude and phase information. As noted in our response to W3&Q1 above, the 2D reshaping serves as just the first step to facilitate convolutional modeling. Subsequently, we employ a multi-dilated convolution architecture to ensure comprehensive consideration of all variables, as detailed in Section 3.2 of the paper. By applying depth-wise 2D convolutions along the 2D variable grid, the model effectively captures interactions among variables at the same frequency components. Specifically, each depth-wise convolution kernel models relationships between variables associated with the same amplitude/phase component.

---

### Official Review · Reviewer_hCV9 · 2025-10-31

**Soundness:** 3
**Presentation:** 3
**Contribution:** 2
**Rating:** 4
**Confidence:** 3

**Summary:**

This paper proposes FACT, a CNN-based architecture for multivariate time-series forecasting that explicitly models fine-grained inter-variable interactions in both the time and frequency domains. The core module, DConvBlock, uses multi-dilated depth-wise Inception-style 2D convolutions to capture variable interactions at each granularity while keeping computation low. Variables are reshaped from 1D into a 2D grid to enlarge the receptive field with fewer layers; frequency modeling is performed in amplitude–phase form and fused with the time-domain pathway via a learnable weight. Across 12 benchmarks, FACT reports state-of-the-art accuracy and notably reduced training time and memory versus attention, especially on high-dimensional datasets.

**Strengths:**

1. The proposed method has valid design—depth-wise, multi-dilated 2D convolutions per granularity plus an amplitude–phase frequency path, fused with learnable weight. The 2D reshape argument is analytically motivated (layer-count reduction).
2. The proposed method shows consistent gains on extensive benchmarks; attention comparison shows lower computations on large datasets; ablations isolate contributions of Inception and FFN and the value of dual-domain modeling.
3. Contributions are explicitly listed; the pipeline and fusion formulae are precise. Efficiency improvements matter for high-dimensional multivariate forecasting, making the approach practical.

**Weaknesses:**

1. The DFT is applied to the embedding dimension (not clearly the temporal axis). The physical meaning and potential pitfalls (e.g., phase wrapping, noise sensitivity, leakage) are not deeply analyzed. A comparison to performing frequency modeling strictly along time (per variable) would help.
2. Reshaping C variables to an H×W grid makes spatial adjacency arbitrary; while dilations broaden coverage, different permutations/layouts might change outcomes. A learned layout or permutation-invariant design could mitigate inductive-bias risks.
3. The components (depth-wise/dilated convs, Inception-style branching, frequency cues) are each established; the paper’s novelty is primarily in the combination and the dual-domain fine-grained framing. A clearer positioning against ModernTCN/frequency-aware models would help.

**Questions:**

1. Is the DFT computed along the hidden/embedding dimension D or along time T? If along D, how should we interpret frequency content in that space, and did you compare to a time-axis DFT per variable?
2. How are variables mapped to the H×W grid (fixed order, padding strategy)? Did you try learned permutations or multiple random layouts with ensembling? Any evidence that adjacency choices matter?

---

> ### Author Response · Authors · 2025-11-24
>
> We have addressed all your comments below with experiments. Thank you for the constructive comments.
>
> ### **W1&Q1. Comparison of applying DFT on embedding v.s., directly on time-axis**
>
> **Response:** Given a multivariate time series $\mathbf{X} \in \mathbb{R}^{C \times T}$ consisting of $C$ variables, we first project each length-$T$ variable into a high-dimensional embedding space of dimension $D$ to get its length-$D$ embedding, resulting in $  \mathbf{X_e} \in  \mathbb{R}^{C \times D}$. We then apply the Discrete Fourier Transform (DFT) to $\mathbf{X_e}$ along the embedding dimension $D$.
>
> It is important to note that the embedding is computed on a per-variable basis; specifically, each $T$-length variable is projected into a $D$-length embedding vector, ensuring that temporal information is implicitly preserved within the embedding space. Compared to applying DFT directly on $\mathbf{X}$ along the temporal dimension $T$, our approach introduces a learnable projection step. This provides the flexibility to adapt to complex temporal patterns within a high-dimensional latent space, yielding more expressive representations for subsequent frequency modeling. As suggested, we compared FACT against a variant that applies DFT directly along the time dimension $T$ of the original series $\mathbf{X}$. The results, summarized in **Table 1**, show that FACT consistently outperforms the direct DFT variant across all datasets and metrics, validating the effectiveness of our design choice.
>
> **Table 1. Comparison of our method FACT and variant applying DFT directly**
> |Model|metric|PEMS03|PEMS04|PEMS08|METR-LA|Traffic2|ECL|Traffic|Solar|
> |---|---|---|---|---|---|---|---|---|---|
> |FACT|MSE|**0.086**|**0.087**|**0.107**|**0.704**|**0.204**|**0.157**|**0.425**|**0.196**|
> ||MAE|**0.194**|**0.194**|**0.200**|**0.456**|**0.179**|**0.254**|**0.276**|**0.254**|
> |DFT directly|MSE|0.101|0.102|0.143|0.724|0.216|0.177|0.467|0.203|
> ||MAE|0.213|0.216|0.236|0.475|0.200|0.274|0.312|0.269|

---

> ### Author Response · Authors · 2025-11-24
>
> ### **W2&Q2. How are C variables mapped to H×W grid? Comparison of different permutations/layouts to get the grid.**
>
> **Response:** We clarify that our method does not assume any prior order among the $C$ variables when reshaping them into an $H \times W$ grid. This work focuses on time series data, regardless of whether the variables have inherent physical order or not. Our method is broadly applicable to various multivariate time series forecasting without being constrained by the need for spatial information. It is beyond the scope of the work to consider such spatial information.
>
> For a multivariate time series with $C$ variables, we reshape the input into a $2$D grid of size $H \times W$ using row-major order, preserving the original sequence of variables as provided in the input. Concretely, the first $W$ variables are placed in the first row, the next $W$ in the second row, and so on. If $C$ is not divisible by $W$, we pad the last row by repeating the first few variables from the beginning of the sequence to complete the $H \times W$ grid.
>
> As suggested, we have conducted experiments to evaluate the impact of different permutations on model performance, which indeed shows that our approach is robust to the choice of variable permutations to grid layout. Specifically, we compare with the Random Perturbation, which randomly shuffles the order of the $C$ variables before reshaping them into the $H \times W$ grid, and the Multiple Random Layouts with Ensembling, which generates several different random shuffling orders and ensembles the outputs of the corresponding models. Since it is unclear and beyond the scope on how to effectively learn a permutation, we do not compare the learnable permutation. **Table 2** below summarizes the results, which indicate that different permutations have minimal impact on the model performance, validating that our method does not rely on specific variable adjacency, and we employ all the proposed techniques to work together for effective forecasting.
>
> **Table 2. Comparison with different permutations**
> |Model|metric|PEMS03|PEMS04|PEMS08|METR-LA|Traffic2|ECL|Traffic|Solar|
> |---|---|---|---|---|---|---|---|---|---|
> |FACT|MSE|0.086|0.087|0.107|0.704|0.204|0.157|0.425|0.196|
> ||MAE|0.194|0.194|0.200|0.456|0.179|0.254|0.276|0.254|
> |Random|MSE|0.086|0.086|0.108|0.725|0.206|0.158|0.426|0.197|
> ||MAE|0.195|0.194|0.202|0.465|0.180|0.255|0.279|0.256|
> |Multiple random with ensembling|MSE|0.087|0.085|0.109|0.715|0.196|0.165|0.433|0.199|
> ||MAE|0.194|0.188|0.204|0.457|0.178|0.260|0.277|0.265|
>
> Additionally, we evaluated the impact of different padding strategies when $C$ is not divisible by $W$, with results summarized in **Table 3** below. Specifically, we compared Latter-padding, which pads the grid by repeating the last few variables, and Random-padding, which pads by randomly selecting variables from the original $C$ variables. The results demonstrate that these padding methods yield comparable performance, indicating that the choice of padding strategy does not significantly affect model effectiveness in this setting.
>
> **Table 3. Comparison of different padding methods**
> |Model|metric|PEMS03|PEMS04|PEMS08|METR-LA|Traffic2|ECL|Traffic|Solar|
> |---|---|---|---|---|---|---|---|---|---|
> |FACT|MSE|0.086|0.087|0.107|0.704|0.204|0.157|0.425|0.196|
> ||MAE|0.194|0.194|0.200|0.456|0.179|0.254|0.276|0.254|
> |Latter-padding|MSE|0.086|0.086|0.106|0.709|0.205|0.158|0.428|0.197|
> ||MAE|0.194|0.194|0.202|0.457|0.179|0.255|0.276|0.256|
> |Random-padding|MSE|0.087|0.087|0.105|0.707|0.203|0.158|0.427|0.198|
> ||MAE|0.195|0.195|0.199|0.457|0.180|0.256|0.277|0.257|

---

> ### Author Response · Authors · 2025-11-24
>
> ### **W3 Clearer positioning against ModernTCN/frequency-aware models**
> **Response:** As suggested, we further clarify our distinctions from ModernTCN and frequency-aware models, and highlight our technical contributions.
>
> As discussed in the introduction, complex variable interactions in multivariate time series often vary across different granularities over time (see Figure 1). Most existing methods primarily capture coarse-grained variable correlations, overlooking finer and dynamic aspects. (i) Therefore, we propose FACT that explicitly models fine-grained variable interactions from both the time and frequency domains at multiple granularities. This differs from ModernTCN, which uses a ConvFFN module (MLP-based feedforward network) that treats each variable as an indivisible whole and mixes information across all variables. FACT also differs from frequency-aware models, which mainly capture temporal dependencies within each variable's frequency components, neglecting interactions between variables at each frequency.  (ii) To this end, FACT introduces a depth-wise convolution block (DConvBlock) that employs depth-wise convolutions with specific kernels to explicitly model variable interactions at multiple granularities. FACT utilizes a stack of DConvBlocks progressively capturing interactions. Differently, in ModernTCN, convolution kernels are primarily used to capture temporal dependencies.  (iii) Furthermore, we reshape the 1D variable into 2D space and design multi-dilated convolution kernels with progressively increasing dilation rates. This design achieves a broader receptive field to effectively capture interactions among all variables in different granularities. In contrast, ModernTCN typically uses large kernels (kernel size 71) to capture temporal dependencies, rather than complex across-variable interactions.  (iv) Extensive experiments demonstrate that FACT's modeling of dynamic variable interactions at fine-grained levels leads to improved performance compared to other baselines, including ModernTCN and frequency-aware models (see Table 1, Table 2, and Figure 12 in the paper). Additionally, depth-wise multi-dilation convolutions result in higher efficiency compared to other baselines, including ModernTCN  as shown in Figure 15 of the paper.

---

> ### Comment · Reviewer_hCV9 · 2025-11-27
>
> Thank you for the response and the new experiments. Especially Tables 2 and 3 have addressed my concerns, and I will increase my score accordingly. (Edit: The edit button is currently inactive; I will make the change during the rebuttal.)

---

> > ### Author Response · Authors · 2025-11-28
> > **Thank you for your effort and for increasing score.**
> >
> > Dear Reviewer hCV9,
> >
> > We are pleased that our response and new experiments have addressed your concerns and that you will increase your score. We also appreciate your great effort and engagement throughout this process.
> >
> > Best,
> > Authors

---

### Author Response · Authors · 2025-12-02
**Rebuttal Summary to AC**

Dear ACs,

We emphasize that **prior to** the OpenReview leakage incident, our rebuttal had already led to a successful outcome, with reviewer ratings raised (**from 4/6/4/6 to 6/8/4/6, with two scores *already* increased and one more reviewer agreeing to raise score, which could have resulted in 6/8/6/6 or higher if not for the incident**). We thank all reviewers for their efforts and hope that the contributions of reviewers, ACs, and authors are recognized.
- Reviewer 5JYU increased score from 4 to **6** on November 25, 00:58 AM AoE before the leakage;
- Reviewer 4AVh increased score from 6 to **8** on November 26, 14:09 PM AoE before the leakage;
- Reviewer hCV9 *agreed to increase score* from 4 to a **higher value** on November 27, 10:35 AM AoE, as our rebuttal addressed the concerns.
- Gratefully, Reviewer 9MyL has a *positive* score of **6** for our work, and did not have time to discuss yet.


Our contributions are as follows (mostly quoting from the reviews):
- **Novel Method with Compelling Ideas on a Valid Problem.** We propose FACT, a novel fine-grained across-variable convolution architecture (9MyL) addressing the timely and important problem of multivariate time series forecasting (5JYU). Unlike existing models that primarily capture macro-level relationships of variables, FACT features a *valid design* (hCV9) and introduces the compelling idea of modeling dynamic interactions at a fine-grained granularity—at each time step and each frequency component (4AVh). The pipeline and fusion formulations of FACT are *precise* (hCV9). The core design enables dependency modeling across both time and frequency domains and flexible receptive fields for capturing dependencies (5JYU). The proposed techniques are *technically novel and effective*, and can be *generalized* to other models (4AVh).

- **Effective, Efficient, and Novel Design of Proposed Techniques.** We introduce the following effective and novel techniques:
    1. We design a depth-wise convolution block (DConvBlock) that leverages channel-specific kernels to model dynamic variable interactions at each granularity. This is regarded as a clever design (9MyL). Our approach to frequency-domain modeling—separating it into Amplitude and Phase rather than just Real/Imaginary parts—is impressive and notable (4AVh, hCV9).
    2. We reconfigure the original one-dimensional variables into a two-dimensional space, reducing variable distance and the required model layers to enable efficient 2D convolutions (hCV9, 9MyL).
    3. We incorporate well-motivated multi-dilated 2D convolutions with progressively increasing dilation rates, enabling fine-grained and dynamic variable interactions in a dual-domain modeling approach that combines time and frequency domains (9MyL, hCV9). The dual-domain architecture is robust (4AVh).
    4. Our designs efficiently achieve a global receptive field across all variables with fewer layers (4AVh). Compared to the $O(N^2)$ complexity of attention-based models (where $N$ is the number of variables), the proposed convolutional architecture—especially after 2D reconstruction—offers substantial advantages for high-dimensional variables (9MyL).

- **State-of-the-Art Experimental Performance.** In extensive experiments across 12 benchmarks, FACT achieves state-of-the-art accuracy and significantly reduces training time and memory usage compared to attention-based models, especially on high-dimensional datasets. This demonstrates high practical value and broad appeal (hCV9, 9MyL), representing a strong contribution (9MyL). The authors present detailed ablation studies that validate the necessity of key model components (e.g., dual-domain modeling, multi-dilated convolution) (9MyL, 4AVh). The reported reductions in training time and memory consumption (up to 50%) are highly attractive practical features (9MyL).

- Besides, our paper is easy to follow, with graphs to show the model & module designs, and visualised results for straightforward comparison and analysis (5JYU).

In our rebuttal, we have conducted experiments and analyses to address all reviewer comments:

- Compared applying DFT on embeddings versus directly on the time axis (hCV9).
- Clarified the mapping of $C$ variables to an $H \times W$ grid, and evaluated different permutations, layouts, and padding strategies (hCV9).
- Compared with alternative reshaping methods and assessed the impact of variable order permutations (9MyL, 4AVh, 5JYU).
- Compared against additional GNN baselines (9MyL).
- Provided visualizations to enhance interpretability (9MyL, 4AVh).
- Performed experimental analyses on the number of layers (4AVh, 5JYU), parameter count and FLOPs (4AVh), and evaluated integration with a SENet block (4AVh).
- Reported performance on specific target forecasting horizons $t+H$ (5JYU).
- Clarified the rationale and positioning of our techniques (9MyL, 5JYU), and the contribution of our work (hCV9).

Thank you for your effort.

Best,

Authors

---

### Meta-Review · Area_Chair_T9TZ · 2026-01-02

**Summary:**

The reviewers raised a series of core concerns that inform the final decision, including comparisons of the performance of other variants of the proposed techniques, adding more experiments and visualizations, clarifying the rationale and positioning of the techniques.

**Reviewer Concerns:**

All key reviewer concerns were addressed by the rebuttal. Specifically, the authors :
* Compared applying DFT on embeddings versus directly on the time axis (hCV9).
* Clarified the mapping of $C$ variables to an $H\times W$ grid, and evaluated different permutations, layouts, and padding strategies (hCV9).
* Compared with alternative reshaping methods and assessed the impact of variable order permutations (9MyL, 4AVh, 5JYU).
* Compared against additional GNN baselines (9MyL).
* Provided visualizations to enhance interpretability (9MyL, 4AVh).
* Performed experimental analyses on the number of layers (4AVh, 5JYU), parameter count and FLOPs (4AVh), and evaluated integration with a SENet block (4AVh).
* Reported performance on specific target forecasting horizons $t + H$ (5JYU).
* Clarified the rationale and positioning of our techniques (9MyL, 5JYU), and the contribution of our work (hCV9).

No concerns remain outstanding.

**Reviewer Scores:**

* Reviewer 5JYU increased score from 4 to 6 on November 25, 00:58 AM AoE before the leakage;
* Reviewer 4AVh increased score from 6 to 8 on November 26, 14:09 PM AoE before the leakage;
* Reviewer hCV9 agreed to increase the score from 4 to a higher value on November 27, 10:35 AM AoE, as the rebuttal addressed the concerns.
* Reviewer 9MyL has a positive score of 6 for the work.

---

### Decision · Program_Chairs · 2026-01-26

Accept (Poster)